# Crucial Role of Central Nervous System as a Viral Anatomical Compartment for HIV-1 Infection

**DOI:** 10.3390/microorganisms9122537

**Published:** 2021-12-08

**Authors:** Ana Borrajo, Valentina Svicher, Romina Salpini, Michele Pellegrino, Stefano Aquaro

**Affiliations:** 1Department of Experimental Medicine, University of Rome Tor Vergata, 00133 Rome, Italy; valentina.svicher@uniroma2.it (V.S.); rsalpini@yahoo.it (R.S.); 2Department of Microbiology and Parasitology, Faculty of Pharmacy, Complutense University of Madrid, 28040 Madrid, Spain; 3Department of Pharmacy, Health and Nutritional Sciences, University of Calabria, 87036 Rende, Italy; michele.pellegrino@unical.it (M.P.); aquaro@uniroma2.it (S.A.)

**Keywords:** human immunodeficiency virus, HAND, reservoir, central nervous system

## Abstract

The chronic infection established by the human immunodeficiency virus 1 (HIV-1) produces serious CD4+ T cell immunodeficiency despite the decrease in HIV-1 ribonucleic acid (RNA) levels and the raised life expectancy of people living with HIV-1 (PLWH) through treatment with combined antiretroviral therapies (cART). HIV-1 enters the central nervous system (CNS), where perivascular macrophages and microglia are infected. Serious neurodegenerative symptoms related to HIV-associated neurocognitive disorders (HAND) are produced by infection of the CNS. Despite advances in the treatment of this infection, HAND significantly contribute to morbidity and mortality globally. The pathogenesis and the role of inflammation in HAND are still incompletely understood. Principally, growing evidence shows that the CNS is an anatomical reservoir for viral infection and replication, and that its compartmentalization can trigger the evolution of neurological damage and thus make virus eradication more difficult. In this review, important concepts for understanding HAND and neuropathogenesis as well as the viral proteins involved in the CNS as an anatomical reservoir for HIV infection are discussed. In addition, an overview of the recent advancements towards therapeutic strategies for the treatment of HAND is presented. Further neurological research is needed to address neurodegenerative difficulties in people living with HIV, specifically regarding CNS viral reservoirs and their effects on eradication.

## 1. Introduction

Infection provoked by HIV-1 is marked by an accelerated reduction in CD4+ T cells, T cell dysfunction, thymic dysfunction, lymphoid destruction, and pan-cellular defects attributed to stem cell dysfunction. In most PLWH, the ultimate result of this immune dysfunction is the development of opportunistic infections and malignancies associated with AIDS. Immune responses can be evaded by this virus by different mechanisms such as the development of permanent infection within diverse types of cells such as monocyte-derived macrophages (MDMs) and T lymphocytes [1].

The diverse origins (embryonic yolk sac, fetal liver, and/or bone marrow) of MDMs have been revealed. MDMs are a cell type generated from peripheral blood monocytes and are widely used to model macrophages for in vitro studies. Due to the importance of the different types of macrophages, regardless of their origin, in HIV-1 infection, in vitro models have always been widely used to examine MDMs.

A key cell reservoir constituted by MDMs can continue to live for a long time despite virus cytopathic events (these data were presented in a recent important study with SIV-infected Asian macaques) [2]. In addition, MDMs can distribute in diverse anatomical compartments (such as the brain) and provide for virus dissemination in the body of PWLH [3,4,5].

The central nervous system (CNS) symbolizes a crucial sanctuary for HIV-1 [6,7]. Resident macrophages support the productive infection of HIV-1 [7] through the blood–brain barrier (BBB) (as shown in in vitro and in vivo studies) [6,7]. 

Minimal penetration of the BBB and probable damage by the toxicity of several drugs render the eradication of HIV-1 reservoirs in the CNS impossible [8,9,10,11,12]. Finding a cure for HIV-1 infection is arduous work since there are many challenges still to be overcome, such as the numerous viral mutations and latency for long periods by integrating into the host genome.

Within the CNS, the virus does not infect neuronal cells; it can establish an infection in different types of cells [13]. Perivascular macrophages, microglial cells, and possibly astrocytic cells produce viral proteins that cause inflammation and, in turn, lead to HIV-1-induced neuronal damage [14].

Viral variants within the CNS undergo genetic compartmentalization that is highly dependent on the various adaptive genetic changes that occur due to specific target cells, and in response to immune selection pressures within the brain microenvironment [6,7].

HIV-associated neurological disease can produce significant morbidity and mortality globally and can be due to HIV replication, opportunistic infections, or comorbidities [15] that can structurally and functionally affect the brain [16]; this affection can occur in a primary or secondary way, also affecting meninges and muscles, among others [17]. HIV-1 infection can produce neurocognitive damage and motor disorders [7]. This infection can be considered as a mix of virus-related neurocognitive diseases and neuronal tissue inflammation [7,18]. The use of cART with an improved ability to penetrate the BBB has drastically diminished the occurrence of these problems [18,19]. Nevertheless, since not all anti-HIV drugs are capable of crossing the BBB with effectiveness, viral reservoirs still persist, causing neurological disorders [20]. As a result, these clinical symptoms still persist as crucial problems for people living with HIV (PLWH), particularly for children or patients with low adherence to treatment [21], the latter becoming increasingly frequent among old PLWH since age is a risk factor for functional impairment and disability [22]. Other crucial risk factors are low levels of CD4+ T cells, an increase in plasma viral load, hepatitis C virus (HCV) coinfection, and metabolic comorbidities in PLWH with neurocognitive impairment [7,19]. Taking into account the restrictions of current therapeutic strategies, novel approaches are a key point in the neuro-HIV research. Some of these methods involve gene therapy, gene editing, RNA interference, and modulation of different cellular physiological processes (reviewed in Ojha et al., 2017, and Kwarteng et al., 2017) [23,24]. Even though the precise etiology of HIV-associated neurocognitive disorders (HAND) in the cART era has not yet been deeply studied, permanent inflammation in the CNS is a frequent characteristic of this disease that can lead to neurocognitive impairment [25].

This review is focused on HAND and the role of the CNS as a reservoir for this virus in the brain. Particular attention is dedicated to crucial topics such as compartmentalization of HIV infection in the CNS and the current advances in therapeutic interventions for the treatment of HAND.

## 2. HAND

### 2.1. Advances and Evolution of the Antiretroviral Treatment of HIV Infection

cART have improved the prognosis of HIV infection, changing the survival of PLWH. As a consequence, immune reconstitution and opportunistic diseases have become infrequent; nevertheless, cognitive disorders in PLWH still persist [20]. 

Neurologic complications of HIV infection are unfortunately common, even in the era of effective cART. Successful virologic control has extended the life span of PLWH, but it is necessary to manage the long-term complications of HIV disease, such as neurocognitive disorders and peripheral neuropathy [26]. The possibility to have powerful and affordable virological and immunological parameters can be useful in addressing these problems by providing data about the burden of HIV-1 reservoirs produced by viral replication in diverse anatomical compartments, as well as on permanent inflammation, immune activation, and senescence in spite of favorable virological clearance [26]. For all these reasons, it is important to provide new insights into the assessment and examination of the disease from a virological, immunological, and clinical point of view [26]. The crucial function of novel markers (total HIV-1 DNA, residual viremia, and immunological parameters) has been deeply studied to optimize therapeutic approaches, improve pharmacological adherence, and individualize examination [26].

Normally, altered neuroimaging findings and the impact of the disease are more frequently observed in older patients [16,27] since studies have revealed the effect of HIV-1 on the neurocognitive situation of patients and brain aging, and on the brain at a structural and functional level [28]. There has been evidence, especially in neuroimaging studies, that older HIV-infected patients have significantly more brain atrophy than their HIV-negative peers [29,30]. In addition, neuronal studies have shown neuronal and cognitive damage, and neurodegeneration and memory failure, and numerous patients with HIV who, in senescence, go from being asymptomatic to developing HAND have been observed [31]. PLWH develop and show this infection in different ways depending on the degree the disease effects the health of each individual. At the moment, no specific cure for HAND has been found. Thus far, there have been small trials showing improvement in neurocognitive function that have been conducted in patients undergoing cART, while intranasal insulin trials are ongoing after in vitro studies showed the possibility of neuroprotective effects of insulin in this disease [32] (Table 1).

An important study provided evidence that paroxetine reduced neuronal cell death produced by proteins of HIV-1 in vitro and in vivo [33]. It has been shown that paroxetine prevented Tat-induced iNOS and inflammatory cytokine expression and Ca2+-induced mitochondrial inflammation and attenuated KCl-induced calcium responses. Additionally, data from this study suggest that the neuroprotective properties of paroxetine in neuronal apoptosis (seen in the brains of PLWH with HAND) can be seen at concentrations of 0.5 to 10 μM. Comparable concentrations help to improve the propagation of neural progenitor cells and can be quickly achieved in the CNS at therapeutic doses since the absorption and enrichment of these components in the CNS are more favorable with 10–20-fold higher dosages in the brain compared to human plasma [33]. It takes weeks or months to reach steady-state levels [33]. These data have also been demonstrated in in vivo studies focused on the neuroprotection and proliferation of neural progenitor cells after systemic administration of paroxetine. Therefore, treatment with paroxetine can be an effective neurorestorative method for ameliorating neuronal death as well as abnormal neurogenesis in HIV-infected individuals with neurologic dysfunction [33]. 

Maraviroc’s good anti-inflammatory effects, high efficacy, and good ability to penetrate to the CNS [34] make it a good antiretroviral in all cell types, including MDMs, highlighting its potential for therapeutic use. Recent longitudinal research employed a randomized controlled design, longer follow-up, and optimized neuropsychological strategies, and this study supported and extended novel observations of improved neurocognition of PLWH with neurocognitive damage who were subjected to maraviroc intensification for 24 weeks [34]. This study also reported partial reversal of monocyte-mediated pathological symptoms previously associated with cognitive damage, namely, a decrease in the percentage of CD14+ HIV DNA and CD16-expressing monocytes and in proinflammatory biomarker sCD163 levels in plasma [34]. 

Kim et al., 2019 studied the systems by which neuronal damage occurs and showed that therapy with insulin demonstrated restoration of dendritic arbors and memory and significant survival of neurons; additionally, full functionality was recovered, and was similar to reversion of memory and synaptic deficiencies in amyloid β precursor protein (APP)-transgenic mice, that were treated with insulin-like growth factor-2 (IGF-2) [35], and to temporary deficit of microtubule-associated protein 2 (MAP2) in dendritic staining of later traumatic brain injury [35]. Important studies have postulated that the mitogen-activated protein kinase/extracellular signal-regulated kinase (MEK-ERK) and CaMKII pathways [35] provide dendritic structural stability with a reversible MAP2–microtubule association. In particular, this system is composed of CaMKII and its regulator NRGN which, in turn, are transcriptionally downregulated in the infection, due to the neurotrophin BDNF, which improves dendrite formation [35], and re-established at normal expression levels with insulin therapy.

Evolution of ratified markers and enriched clinical cognitive analysis that can holistically and correctly evaluate the possibility of evolving HAND are also needed to promote future trials of novel HAND strategies.

Additionally, the role of different parameters (total HIV-1 DNA, residual viremia, immunological biomarkers) has been the focus of optimized therapies, enhanced pharmacological adherence, and individual examination [26].

### 2.2. Classification of HAND

Classification systems of HAND are necessary for the diagnosis of this infection. To obtain a reliable diagnosis, we need to follow some crucial and standard criteria: the Frascati criteria [36]. These are described according to different levels of severity of the disease, such as asymptomatic neurocognitive impairment (ANI), mild neurocognitive disorders (MND), and HIV-associated dementia (HAD), which is the level with the highest severity in HAND [7]. ANI includes at least two neurocognitive domains that do not impede in the patient’s daily activity, do not follow established criteria for dementia, do not provide proof of another pre-existing cause of ANI, and include very mild difficulties (also at the motor level) [7]. The early detection and diagnosis of ANI are very important clinically since it can rapidly progress to HAD.

Regarding MND, they represent a HAND level that implies at least two capacity domains that slightly interfere in the daily function of patients and some acquired damage in neurocognitive functions. To assess the neuropsychological status of PLWH, the Memorial Sloan Kettering (MSK) scale of 0.5 to 1.0 is used. This scale takes into account the following skills: level of mental acuity and information processing speed, verbal/language skills, abstraction/executive functioning, efficiency and attention in working memory (learning, memory), homemaking, sensory-perceptual skills, motor abilities, and social functioning of HIV-1 patients; neurocognitive damage does not follow established criteria for dementia, and it does not meet criteria for delirium [7]. 

The highest level of severity of the disease corresponds to HAD, which is characterized by significant acquired neurocognitive damage and involves multiple skill domains (at least two), and this causes difficulties in carrying out daily activities (work, life in the workplace, home, social activities) and functional learning of them, and very low attention and concentration to process new information [7]; the pattern of neurocognitive impairment produced by HAD may not follow the criteria for delirium, and although it exists, an MSK scale should be established for dementia when delirium is not present. In PLWH who are drug addicts or who have severe episodes of depression with significant functional difficulties and/or psychosis, the diagnosis of the disease should be made when the patient is not in a depressed state or after 1 month of having stopped using drugs [36].

### 2.3. Pathology of NeuroAIDS

The pathogenesis of HAND itself is complicated and intricate, and it has been revealed in recent studies that functional alterations in neurons were associated with the pathophysiology of HAND. 

Productive HIV infection occurs in perivascular macrophages, MDMs, and microglia [34]. While it is sufficiently established that neuronal injury and loss are correlated with the evolution of HAND manifestations, there is a paucity of available information on the capability of HIV to infect neurons [13,35]. In accordance with the broadly accepted model, in a systemic infection, after at least 1 week, the virus enters the CNS through the BBB, infecting it via a “Trojan horse” pattern [36] (Figure 1). HIV-1 uses infected lymphocyte cells, CD4 + T cells, and monocyte cells, which later differentiate into macrophages [35], to infiltrate, infect, and activate cells that are in direct contact with perivascular macrophages and astrocytic and microglial cells; subsequently, they transmigrate to the perivascular space of the CNS while evading immune detection [37]. Although astrocytes are sensitive to HIV infection, they do not promote productive infection; conversely, perivascular macrophages and microglia are the only cells in the CNS able to support HIV infection in the brain [13,36] (Figure 1). In the setting of HAND, symptoms are correlated with loss of neurons and cellular damage. The importance of secretion by activated microglial cells, macrophages, and astrocytic cells of small metabolite chemokines, inflammatory cytokines, and neurotoxic viral proteins has previously been demonstrated, and they can cause significant neuronal damage and disrupt the BBB, producing a continuous viral influx [38].

A controversial question concerns the ability of HIV to produce infection in neuronal cells, even at low levels, and a large number of studies, since the 1980s and 1990s, have reported the capability of HIV to produce infection in neuronal cells in the CNS in vivo [38]. Important studies utilizing in situ PCR and immunohistochemistry reported the existence of HIV genetic material and antigens in neuronal cells [38]. Additional research isolating neurons from autopsy brain tissues of PLWH utilizing laser capture microdissection (LCM) demonstrated the presence of HIV pro-viral DNA in neuronal cells by PCR [39,40]. Another study used hyperbranched multi-displacement strategies for whole-gene amplification via PCR and analyzed the existence of HIV DNA in neuronal cells from post-mortem brain tissue obtained by LCM [41]. In vitro studies also demonstrated the possibility that HIV-1 could infect human neuronal cells [42]. Nevertheless, authentication of pathologically significant infection of human neurons in vivo remains to be demonstrated. In addition to its existence in the CNS of PLWH, HIV has also been discovered in the CNS and in the developing fetal brain of infected pediatric patients [43].

Sturdevant et al. postulated that HIV-1 likely enters the CSF/CNS at low levels via partial partitioning of viral particles through the BBB, or via background levels of trafficking of immune cells and small numbers of infected CD4+ T cells. Often, in the process of the primary infection, HIV-1 viral RNA levels are elevated within the CNS, and this can be attributed to higher levels of CSF pleocytosis [44]. Thus, the early CNS viral burden may result from the release of the virus from raised numbers of infected CD4+ T cells trafficking into the CNS, responding to local HIV-1 replication in the CNS or another inflammatory condition [45].

There are mainly two models, direct and indirect, to explain the degeneration and development of neurological symptoms in HAND.

The theory postulated by the direct model is that infected cells infiltrate the CNS, secreting viral proteins, and produce neuronal impairment through direct connection with neurons [13,37]. The indirect theory propounds that neuron injury is arbitrated by the inflammatory response provoked by infected and uninfected glial cells within the CNS against viral infection and HIV proteins released by directly infected cells [13,37]. HIV-1 can lead to an inflammatory response produced by the secretion of viral proteins (viral surface glycoprotein 120 (gp120), transactivator of transcription (Tat), and viral protein R (Vpr)), several soluble molecules and cellular products (such as quinolinic and arachidonate acids, adenosine triphosphate (ATP), platelet-activating factor (PAF), excitatory amino acids, superoxide anions, matrix metalloproteases, growth factors, and proinflammatory cytokines such as tumor necrosis factor-alpha (TNF-α), interleukin-6 (IL-6) and interleukin-1 (IL-1), C-C Motif Chemokine Ligand 2 (CCL2), and Regulated upon Activation, Normal T cell Expressed and Secreted protein (RANTES)) [13,44], and glutamate and nitric oxide (NO) radicals secreted by astrocytic cells, causing their cell death [26,46,47]. There are some molecules that have a neuroprotective effect such as growth factors and β-chemokines, and others that have an adverse neurotoxic role such as excitatory amino acids and other agonists of the N-methyl-D-aspartate glutamate receptor (NMDAR), which also reduce glutamate uptake [13,44].

Stimulation of Nod-like receptor pyrin domain containing 3 (NLRP3) inflammasomes in microglia can be produced by the HIV-1 Tat protein as well as raised caspase-1 and IL-1β levels, which intensify and cause exacerbated inflammation [44]. Moreover, Beclin-1-dependent autophagy activation caused by HIV-1 infection releases the p24 viral protein and cytokines that activate the proinflammation response. 

In addition, HIV-1 infection produces damage and dysfunction of mitochondria and oxidative events, with the release of reactive oxygen species (ROS), reactive nitrogen species (RNS), and inducible hypoxia factor (HIF)-1 in microglial cells [44].

Oxidative events produce stresses that increase neuronal apoptosis, even in the absence of HIV-1 infection. It has been found that oxidative stress has a crucial function in a number of other neurodegenerative diseases [48].

### 2.4. HIV Viral Proteins Involved in Neuropathogenesis

The most profoundly examined and possibly most relevant viral protein is the neurotoxic surface protein gp120; moreover, it is implicated as an important factor in the pathogenesis of HAND. 

When the neuronal *N*-methyl-*D*-aspartate glutamate receptors (NMDARs) are activated, the phenomenon known as excitotoxicity occurs due to the excess of glutamate, which causes an excessive calcium flow, the formation of free radicals (NO), mitochondrial damage, and ROS generation, concomitant with lipid peroxidation and caspase activation, which causes damage to neurons, dendritic degeneration, and apoptosis [44]. 

The gp120 protein causes neuronal injury by activation of *N*-methyl-*D*-aspartate (NMDA)-coupled NMDARs, resulting in unusually high calcium flow, while Tat phosphorylates NMDARs, which potentiates glutamate excitotoxicity [44]. Both Tat and Vpr have also been shown to be involved in neuronal impairment, with neuronal cell death related to Tat [49]. In addition, gp120 promotes oxidative events and dysfunction, generation of ROS, and raised cerebral endothelial permeability [50].

The gp120 protein has many functions, and among them, it has the ability to produce the release of inflammatory cytokines and neurotoxic components [51] that generate neuron damage in the CNS and some downregulation of tight junction proteins [52]. The entry of the virus and the induction of an effective infection in the cells are promoted by the action of different elements such as the presence of CD4, the chemokine receptor CXC 4 (CXCR4), and the chemokine receptors 3 and 5 (CCR3 and CCR5) mediated by protein kinase C (PKC) coreceptors, in addition to processes such as intracellular calcium secretion [52]. The apoptotic processes are influenced by gp120, which causes dysregulation of calcium homoeostasis, stimulation of oxidative stress, and activation of the proapoptotic transcription factor p53 [53].

Previous studies have revealed a destructive and crucial function of HIV-1 Tat in the development and progression of HIV-1 infection [54], and when the disease is manifested, it is one of the first HIV proteins to be expressed. Several studies have revealed that Tat contributes to neuronal impairment in diverse ways. Tat crosses the BBB because of its ability to modify the expression pattern of proteins critical for the integrity of the endothelial tight junctions [52], and to promote leukocyte infiltration and invasion [53]. 

In previous studies, Tat was revealed to have cytotoxic and proinflammatory effects that stimulate microglia; synthesize proteins, molecules, and other factors such as monocyte chemoattractant protein type 1 (MCP-1), adhesion molecules of the CAM protein family (V-CAM 1 and I-CAM1), PAF [54], and free radicals; and interfere with molecular mechanisms controlling cyclic adenosine monophosphate (AMP) levels, intracellular calcium concentration, and ion channel expression [54]. The exposure of human astrocytoma cells to the HIV-1 Tat recombinant protein produced high levels of apoptosis compared to untreated cells [54]. In accordance with this finding, cerebral injection of Tat within mice provoked raised Ca2+ v1.2 channels, producing astrogliosis in the cortical region and successive death of cortical neurons, microglial cells, and MDMs [55].

A previous finding reported that Tat can also produce glial fibrillary acidic protein (GFAP) upregulation, which can provoke GFAP aggregation and the production of ROS in astrocytic cells [56]. The HIV-1 Tat protein also has an important function in upregulating the Cx43 human gene, implicated in the CNS gap junctional communication and inducing, if over-expressed, cell death and inflammation, by binding the Cx43 promoter, and thus raising Cx43 mRNA formation [57]. In vitro and in vivo experiments of diverse studies have reported that Tat provokes a strong excitatory state that may intensify the neurotoxic and excitotoxic activities at the presynaptic level [58]. Tat provokes deficiency of post-dendritic synapses as a result of the interference with glutamatergic signals [59], with the presence, within the striatum and midbrain, of levels of the dopamine transporter [60].

HIV-1 Vpr is released from cells infected by HIV-1. This protein is involved in the apoptosis process, presumably through the production of IL1- and interleukin 8 (IL-8), which promote the release of neurotoxins such as matrix metalloproteinases and induce proteins involved in the cell cycle and proapoptosis [58]. This protein activates the release of proinflammatory cytokines, such as TNF-α, IL-1, and IL-8 in MDMs, and presumably can act on the mitogen-activated protein kinase (MAPK) pathway [58]. 

HIV-1 negative regulatory factor (Nef) is a flexible, multifunctional protein with several cellular targets that is necessary for the pathogenicity of the virus [61]. This protein provokes astroglial cell activation and astrogliosis [62]. Nef favors lysosome permeabilization, stimulating the release of enzymes [63] and the successive cell death of microvascular endothelial cells (MVECs) [64].

Previous reports have also shown that gp120 and Tat were associated with neurotoxicity. The protein gp120 was reported to be toxic to dopamine neuron cultures [64], reducing the capability of neurons to transport dopamine [65]. The toxicity of this protein was first described by Nath et al., 1996 [66].

The full length of Tat has a total length of 86–104 amino acids, and peptide analysis of different overlapping lengths did not yield neurotoxic processes in cultures of primary neurons. After this study, it was investigated whether Tat was toxic to fetal neurons cultures through a calcium-dependent process or by raising oxidative stress. Previous studies utilized direct intra-striatal injections of Tat that result in increased carbonyl formation [67]. Increased gliosis, cellular injury, and neuronal death have been associated with an increase in apoptosis [68]. Alteration of calcium homeostasis, activation of TNF-α, nuclear factor κappa-light-chain-enhancer of activated B cells (NF-κB), and glutamate receptors, and stimulation of NO production are the basis of Tat-mediated neurotoxicity. Similar to Tat, gp120 has been found to provoke neurotoxicity via different pathways. In both in vivo and in vitro experiments, gp120 exposure has been demonstrated to provoke cell death [69]. NMDA decreases gp120-induced toxicity. Stimulation and activation of the NO synthesis pathways have also been shown following the administration of gp120 [69].

The Tat protein interacts with the NMDA receptor that produces high levels of Ca2+/calmodulin-dependent protein kinase II, which reacts with glycogen synthase kinase 3β and causes the death of oligodendrocytes, a reduction in myelin-like membranes in mature oligodendrocytes, and HIV-1 neuropathology [69].

Studies have postulated that the motives of evolution of this infection are multifaceted, as different episodes involve a torrent of pathogenic repercussions. It is an interesting objective to correlate the detected increment in ROS production with the augmented viral load burden in the CNS despite there being several main reasons. Furthermore, patients with late-stage HIV-1 infection have enjoyed benefits from treatment with antioxidant compounds, as these solutions lengthen survival, but they do not stop the ongoing oxidative stress events, either because the phenomena are irreversible or because of the high production of ROS, which may no longer be eliminated by the weakened antioxidant cellular defense processes. Additionally, a specific study stated that because of the extracellular Vpr concentrations that are very low in the early phases of disease compared with the late stages, it is appealing, although speculative, to hypothesize that in the early disease stages, exogenous levels of Vpr are too low to cause immediate detrimental effects, and the observed downstream effects are generated only later during disease progression when Vpr levels rise above this threshold. It is possible, however, that in the early stages, astrocytes, although exposed to subthreshold concentrations of Vpr, might become sensitized to subsequent exposure to Vpr, meaning that lower amounts of Vpr can impair astrocytic functionality [70].

Akiyama et al., 2020 demonstrated that IFN-I-dependent proinflammatory responses in MDMs are generated by HIV icRNA expression alone, although HIV icRNA expression does not generate functional proteins of the virus or virions, such as gp120 and Vpr [71]. Infection with the virus exhibits HIV icRNA nuclear export with Rev mutant deficiency, which does not produce innate immune activation in microglial cells and generates multiple spliced viral RNAs, indicating de novo Tat expression is not the cause of HIV-induced microglia activation. Curiously, HIV icRNA (gag mRNA) has been discovered in the CSF of patients undergoing cART [71], and the precise RNAScope assay has detected the existence of levels of SIV gag mRNA (icRNA)-positive cells in cART-suppressed monkey brains [71]. In addition, these viral icRNA-expressing cells in the CNS (microglial cells) influence the state of neurons in cART-suppressed PLWH and produce inflammatory cytokines. Various cure candidates that eliminate the establishment of HIV icRNA (Rev and Tat inhibitors) [71] might have clinical advantages in suppressing irregular HIV icRNA-induced inflammation and lowering the prevalence of HAND patients undergoing cART [71].

### 2.5. Inflammation and Role of Mononuclear Phagocytes in HAND

In addition to the direct toxicity of HIV proteins, mononuclear phagocytes, including perivascular macrophages and resident microglial cells, have an important impact on the evolution of the most severe level of HAND: HAD.

Inflammation has a crucial function in the production of phenomena that provoke neurodegeneration in this infection. The entry of HIV into the CNS occurs via the BBB, which is carried out by circulating monocytes in response to the chemotactic signals produced within the parenchyma, and the monocytes are responsible for the stability of the disease in perivascular macrophages of the CNS [41], microglia [34], and astrocytes [72].

Zhou et al. showed that direct HIV-1 infection of CNS macrophages is crucial for HAD establishment and evolution [69]. They discovered broad productive HIV infection in PLWH with accelerated progression of disease, with low cellular infiltration, compared to those who evolved more gradually [69]. Infection within macrophages and CD8+ T cells was found to be common in the deeper midline and mesial temporal structures only in patients with HAD, which exerted effects on cognitive damage during the disease [69].

One question that has not been thoroughly studied is why inflammation within the brain is maintained even when the replication of the virus is restricted by cART [73]. On the one hand, one hypothesis supposes that the inflammatory responses, triggered by HIV, stimulate the proteasome to evolve into an immunoproteasome that obstructs the turnover of folded proteins within cells of the CNS and alters cellular homeostasis and response to stress [74], producing severely damaged neuronal and synaptic protein dynamics, in some way conducive to HAD. The result of an important previous study suggested that immunoproteasomes may be implicated in the control of synaptic proteins in this infection. Successively, synaptosome protein changes related to immunoproteasome generation could produce morphological modifications in the synapto-dendritic arbor, which is related to HAND [74]. 

It has been reported that an important component of the immunoproteasome, TRIM5α, was independently silenced utilizing siRNA, and the impact on IFNα-induced viral elimination was examined. The useful inter-dependence of the IFNα-activated anti-HIV-1 phenotypes of TRIM5α in humans and the immunoproteasome was established by the reduction in the levels of IFNα elimination that was shown following PA28A silencing in cells that lacked TRIM5α [75]. In spite of the existence of sporadic data of human TRIM5α that affects this infection either by elimination of certain HLA-associated cytotoxic T lymphocyte (CTL) escape mutant viruses or by producing autophagy in Langerhans cells, these data show non-strain-specific inhibition of HIV-1 infection by human TRIM5α. Principally, this study reported that TRIM5α is functional in CD4+ T cells and is dependent on IFNα and stimulation of the immunoproteasome. The IFN levels are high during the acute and chronic phases of natural infection with HIV-1, and it can be supposed that TRIM5α is partly responsible for the HIV-1 immune control in patients, a result with important associations between positive clinical results and raised TRIM5α expression or specific TRIM5α alleles [75]. 

Additionally, an interesting topic concerns the possible effects of HIV-1 inhibitors on possible immunosurveillance, which depend on the particular role of proteasome and CTL epitope generation as well as on the relative improvements in the CTL response as an antagonist to the recovery of CD4+ lymphocyte cells and antibody production [76] in immunosurveillance. Antiviral drugs with proteasome inhibitory capacity, such as ritonavir/saquinavir, are able to modulate the presentation of Ags to CTLs and may perhaps be exploited further to find new strategies for treatment of autoimmune disease, chronic immunopathologies, or disease caused by transplantation reactions.

On the other hand, another event of permanent CNS inflammation that has been demonstrated is microglial priming from circulating microbial translocation, which results from gut bacteria and a disturbed microbiome. There is evidence that microbial products can transverse to the BBB through different mechanisms. It has also been indicated that CNS inflammation in PLWH undergoing cART could proceed from a debilitated pattern of immune reconstitution inflammatory syndrome [77].

Morbidity and mortality in HIV infection are mainly due to microbial translocation, probably due to the persistent inflammation that it produces and maintains [78,79]. The relationships between disease progression, microbial translocation, and mortality do not depend on viral inhibition by ART in PLWH [80]. Different studies have shown direct correlations between plasma levels of LPS (lipopolysaccharide from the surface of Gram-negative bacterial–microbial translocation products) in residual viremia of PLWH, stimulation of CD38+HLA-DR+ CD8+ T cells, and stimulation of monocytes, interferon-sensitive genes such as MxA, and proinflammatory cytokines such as IFN-α, IL-6, and TNFα.

LPS levels and/or bacterial DNA levels have a direct association with other biomarkers of microbial translocation and innate immune activation, such as soluble CD14 (released by monocytes of bacterial stimulation), LPS-binding protein (LBP), and endotoxin. There are many inflammatory events that occur during HIV infection (virus replication, opportunistic infections, etc.). Studies in which no association with HIV was explored showed that microbial translocation and inflammation are associated. In idiopathic CD4 lymphocytopenia (ICL), LPS is found at high levels and is related to the proliferation of CD4 + T cells; in the colon in uninfected pig-tailed macaques, LPS levels were related to the interferon-responsive MxA gene in the GI tract [80], which shows that microbial products can directly activate inflammatory responses.

According to a recent study, increased neuropathy can occur due to control of viral replication in the brain and the loss of immune stimulation by the gut microbiota. Therefore, the loss of protective resident microbes can lead to CNS dysfunction [79].

LPSs that are sufficient to prime microglial cells for antigens presented to efficiently eliminate the virus have been implicated in CNS maturation [79,80]. Toll-like receptors (TLRs), expressed by microglial cells, are thought to act only during infection. It has been shown that the microbiota regulates microglial cell function through TLR-4, which prepares these cells to attack an infection. Microglial cells evolve early in embryogenesis from the progenitors of the yolk sac; however, unlike macrophages, microglial cells live long, without any support from circulating blood cells [79]. In addition, Toll Like Receptor 4 (TLR4) exists in the intestinal microbial products that are circulating in the blood and could thus reach the brain [81]. Evidence exists showing that the microglial cell signaling of TLR4 causes stimulation of the microglial cells, and that signals from the gut microbiota can be transferred to the brain from the enteric nervous system [75]. Actually, it has been reported that enteroendocrine cells have neuropods (axon-like basal structures that contain neurofilaments, which are typical structural proteins of axons) that are connected to neurons and are capable of transferring signals to the brain [81]. Brown et al. administered LPS orally, and this administration blocked effects of gastrointestinal exposure, but oronasal–pharyngeal and pneumonic exposure may occur as well. In relation to this topic, dissimilarities detected between feeding mice LPS alone and in combination with a TLR1/2 ligand (Pam3CysK4) indicate that further investigations are necessary to study the interaction between multiple TLR ligands on microglial cell function and response to infection. Furthermore, in this study, it could not be completely ruled out that there is a contribution from cells of gut-resident CX3CR1 to this phenotype or other migrating DC cells [82]. Important studies have indicated that cells of the gut could reach the CNS, and that there is a population of long-lived CX3CR1 cells in the gut [83]. Other studies postulated that disruption of the gut epithelial barrier may allow unregulated translocation of gut microbes into the lamina propria. Thus, factors of bacteria can penetrate the gut-associated lymphoid tissues (GALT) and lumen of the blood, where they communicate with various immune cells and can trigger effector-type T cell differentiation [84]. Regulatory T cells that survey the GALT, blood, and CSF and modifications to the local microbiome can promote T cell brain penetration. Factors of bacteria could upregulate inflammatory cytokine levels, alter BBB integrity, and produce inflammation [80]. Additionally, factors of bacteria can produce LPSs and can regulate endothelial TLRs to produce brain inflammation and infection within the CNS [84,85].

Additional future investigations remain to be executed to examine how gut microbes affect distal CNS events.

The clinical conclusions of these findings reveal that immunostimulatory LPSs are derived from the microbiota, and the structure of the microbiota and the immune stimulatory products being released by the microbiota likely dictate the maturation of microglial cells in the brain. In addition, fatty acids that can be generated by the microbiota are also able to bind TLR4 and might establish other agonists [86,87]. As microglial cells can modify the role of neuron cells, these data propose that microbial control of microglial cells could affect the activity and duty of neurons [88,89].

## 3. The CNS as a Viral Reservoir for HIV Persistence

The CNS is considered as an immune reservoir site in which persistent infections occur, which dampens efforts focused on developing eradication strategies. The different criteria to consider the CNS as a reservoir of HIV-1 include the following: (i) it can incorporate an integrated virus; (ii) HIV-1 is maintained in a latent condition; (iii) and the virus, with significant abundance, is capable of replicating within long-lived cells resident in the CNS [82]. Furthermore, specific proteins of HIV-1 (Tat, Rev, and Nef) can be generated without virion generation [90]. 

Certainly, to achieve a real sterilizing cure, the latent HIV harbored in the brain should be eradicated since it can be re-stimulated and then reseed a systemic infection. In vivo, HIV dissemination in the CNS occurs in perivascular macrophages and microglial cells [90]. Previous studies have identified viral particles in the brains of PLWH with non-detectable viral load in the plasma and cerebrospinal fluid (CSF) [91]. Moreover, it has been shown that viruses in the CNS have singular long terminal repeat (LTR) promoters which have mutations in the Sp motif next to NF-κB binding sites, which fosters viral latency [92]. Perivascular macrophages, microglia, and astrocytes (target cells in the CNS) have long half-lives that permit the virus to linger within brain cells and allow the preservation of the brain reservoir of HIV-1. Lastly, the recurrence of CNS cell infection has been explored in recent studies that examined macrophages and microglial and astrocytic cells. PLWH without neurocognitive damage presented infection in 17%, 14%, and 11% of macrophages and microglial and astrocytic cells, respectively, and PLWH with injured neurocognitive condition showed infection in 30%, 9%, and 19%, respectively [93]. Then, other studies on infected brain tissues also showed that the frequency of HIV-1 infection is more elevated at or close to blood vessels than in regions further away from the vessels [94,95]

Trials based on raltegravir escalation were characterized by the presence of a low level of HIV within the CNS during suppressive cART. In one report, blood and CSF samples of HIV-1 patients were used, and CSF HIV RNA below 50 copies/mL were re-assessed with a precise single-copy assay. Approximately 76% of patients had detectable plasma HIV RNA, and 6% of HIV-infected persons had detectable CSF HIV RNA, by a single-copy assay [96], showing, in cognitively balanced PLWH, low-level replication of HIV within the CNS, which was undetectable by standard methods. In patients without symptoms, variations between blood and CSF HIV RNA levels have been noticed [96].

Previous in vivo studies based on simian immunodeficiency virus (SIV) models have demonstrated strong confirmation of the persistence of SIV DNA after extended elimination of virus replication with cART [97]; other studies showed the important role of macrophage populations for the productive infection of HIV, with utilization of BrdU labeling and biomarkers for the differentiation of macrophages [98]. 

Numerous research works of HIV-RNA levels in the CSF have also revealed the existence of latent infection in the CNS despite plasma virus elimination being below measurable clinical limits. This event is known as CSF viral escape and aids the role of reservoirs within the brain [97]. This event can happen in 5–10% of cART recipients, is related to immune stimulation and major depressive disorder [99], and most likely derives from local reservoirs of HIV-1 [91]. Good knowledge of CSF viral escape could contribute to obtaining crucial observations about infection in the brain involving the different types of cells in the CNS provoking permanent HIV infection, and could be key to the successful eradication of both latent and productive HIV from the CNS [92]. However, analysis of CSF viral escape is restricted in all contexts by its relatively low prevalence and the requirement for a lumbar puncture for examination [100].

### 3.1. Compartmentalization of HIV within the CNS

The development of compartmentalization of HIV in the CNS starts with divergent development from a systemic virus, produced in genetically unconnected viral strains in the brain [101]. This phenomenon usually starts in primary HIV infection and has also been related to the evolution of cognitive damage [102]. The genetic flow limited to HIV and evolution and divergence from the virus which spreads in the peripheral blood are caused by anatomical sites called compartments. In this way, the virus is protected from determined immune responses, variations in biochemical levels, and different antiretroviral treatments, generating an ideal situation for interactions between the pathogen and host.

The existence of diverse virus populations of HIV in the CNS from patient autopsy samples, such as CFS, brain tissues, or blood [101], has been demonstrated. HIV-1 virus variations in CSF versus blood have been explained by different theories that postulate [102] (Figure 2) three levels of the virus in the CSF compared with the blood: equilibrated (where viral populations in the blood and CSF are analogous), compartmentalized (where the virus in the blood and CSF viral populations are very different, expressing independent evolution of populations in these compartments), and clonal amplification (where a unique variant is largely extended within a compartment) [103] (Figure 2).

Viral compartmentalization, diverse in viral populations at the genetic level, can be identified in the CSF/CNS during the progression of infection, and replication occurs in the periphery [104]. Two important classifications of compartmentalization have been explored in previous studies [105]. The first is clonal amplification, characterized by the fast growth of genetically analogous variants producing a CSF viral population of minor complications. These viral populations demand elevated levels of CD4 for entry and are CCR5 (R5)-using T cell-tropic viruses [106] (Figure 2). Clonally amplified populations have been discovered as early as two to six months post-infection during the primary infection [107]. Another more complicated classification of CSF viral compartmentalization has also been described, frequently made up of macrophage-tropic viruses. These more genetically complex populations frequently coincide with a prolonged time since virus infection, demonstrative of a more extensive stage of isolated replication and development of the population and entry phenotype [108]. 

Regarding the compartmentalized HIV-1 populations, R5 T cell-tropic viruses are clonally amplified and associated with cell death, and macrophage-tropic viruses are associated with a high possibility of having neurocognitive disorders, although macrophage-tropic viruses are clearly genetically diverse, since they are capable of infecting cells with low CD4 expression on the surface. This does not happen with macrophages from healthy donors since they have a high variability [108]. R5 T cell-tropic and macrophage-tropic populations can be seen in the early evolution of viruses: for example, in samples of CFS from infants that use lower levels of CD4 to become macrophage-tropic compartmentalized [108]. The detection of these compartmentalized populations is very important because these can occur in infected patients without signs of cognitive neurodeficiency [109]. One study by Sturdevant et al. revealed compartmentalization of a CSF/CNS lineage of the virus that had been settled less than six months post-infection and remained for at least two years [106]. It has also been demonstrated that under conditions of primary infection, in different moments after infection, diverse compartmentalized populations within the CNS have been found, leading to the belief that even when compartmentalized populations are not conserved, the CNS can produce a propitious environment for infection [109].

The significance of the adequate penetrance of diverse drugs into the brain is a subject of further study [110], and it has been fully corroborated that lower concentrations of antiretroviral drugs in the CNS do not achieve optimal virological suppression. There are few pharmacological studies of cART with respect to tissue macrophages. A more greatly elevated maximal effective concentration (EC_50_) in macrophages than in lymphocyte cells might diminish cART effectiveness in these cells [111]. In addition, a reasonable percentage of astrocytes isolated from post-mortem brain tissues of HIV^+^ patients demonstrated integrated HIV [112]. The capability of HIV-1 to integrate into terminally differentiated astrocytic cells indicates a stable reservoir of a pro-virus in the CNS that affects the evolution and achievements of therapies involved in the eradication of HIV-1 [41,113].

### 3.2. Towards Strategies for the Eradication of CNS Reservoirs

Numerous studies have explored the presence of a unique viral reservoir in the CNS. The CNS poses important challenges for eradication procedures. The first step is to reach the target CNS reservoir during eradication strategies and identify efficient drugs for the stimulation of HIV-1 in non-CNS compartments that can appropriately pass through the BBB [91]. Due to the fact that the CNS is an immune-advantaged compartment, cART does not have effective ingress through the BBB and efficiency within the CNS [114]. Previously, lamivudine, stavudine, and zidovudine acted by favoring the decrease in CNS cells; however, at present, other antiretroviral drugs such as protease inhibitors are an important component of cART, where targeting the CNS sanctuary of HIV with new protease inhibitors capable of crossing the BBB and interrupting HIV in the brain may be key to treating or even preventing HAND [110]. 

Lamivudine, stavudine, and zidovudine are used in neuro-cART regimens, but there are studies that demonstrated that the antiviral action of them is not the optimal strategy to treat HIV-1 CNS infection in different cell types of the CNS, including perivascular macrophages, microglia, and astrocytes. 

The pivotal role of astrocytic cells is critical in functions related to the homeostasis of extracellular K+ and intracellular Ca2+, permeability of the BBB, and clearance of glutamate, and neuro-cART regimens could induce astrocytic cells to alleviate cognitive damage. HIV-1 infection in astrocytic cells can cause astrocytosis, and apoptosis of neuronal cells with gp120-mediated and Tat-mediated neurotoxicity. Both of these HIV-1 proteins are responsible for severe neurotoxicity in the CNS [108]. There is evidence that suggests that Nef is released into neurons by extracellular vesicles and also contributes to the neurotoxicity related to HAND (which, in turn, compromises autophagy) [110].

There are drugs such as maraviroc, darunavir, and raltegravir that are able to block MMP-2 and MMP-9 with non-toxic levels for astrocytes [110]. 

Darunavir is an FDA-approved protease inhibitor, widely used in the treatment of PLWH, with a significantly greater virological response and immunological benefits compared to the standard of care. Ghosh et al. realized that a structure-based modification resulted in very potent brain-penetrating protease inhibitors, namely, GRL-04810 and GRL-05010 [111]. They also designed other inhibitors such as GRL-0739 and GRL-10413, which show BBB efficacy. Data from these studies suggest that GRL-04810, GRL-05010, GRL-0739, and GRL-10413 have several advantages: (a) they form extended connections with the residues in the active site of HIV-1 proteases, leading to excellent antiviral activity against a broad spectrum of drug-resistant HIV-1 isolates and variants; (b) the hydrophobic interactions and logD values indicate a very good lipophilicity profile; and (c) they can efficiently penetrate the BBB and retain their activity [111]. Therefore, they possess desirable features as drugs suitable for treating PLWH with wild-type and/or multidrug-resistant HIV-1 variants. The current findings warrant further consideration of these novel PIs for the therapy of HIV/AIDS. It is necessary to evaluate their other drug properties, including their pharmacokinetics, pharmacodynamics, and oral bioavailability, in the clinical setting [111].

### 3.3. CNS Penetration Effectiveness (CPE) Scoring System 

Antiretroviral drugs have been classified in accordance with a CNS penetration effectiveness (CPE) scoring system, which is a graduate system scored from 1 to 4, with 1 being less advantageous and 4 the most helpful [111] (Table 1). 

Active cART involves cART plans that have a minimum combined CPE score of eight or more [10]. Hence, cART with a CPE score of at least 8 or more must be used to guarantee its effectiveness in the CNS and the impediment of the establishment of de novo infection [111,113]. Gray et al. showed that viruses from different compartments display distinct behaviors and responses to antiretroviral drugs, with brain viruses being less responsive than, for example, blood viruses [109]. In the end, in the choice of an efficient ARV, it is important to guarantee immunity, and to ensure that it is effective in the elimination of any cell that harbors the reactivated virus. In contrast with other body organs, the CNS holds a modified immune system due to its immune-favored condition [92,111]. In summary, the studies explored in this research demonstrate that a cure approach based on antiretroviral drugs, with CNS bioavailability and effectiveness to eradicate the CNS viral reservoirs and implicate the increase in and promotion of the CNS immune system to help viral depuration, is necessary. Nevertheless, this would have to be adjusted with care to prevent CNS immune reconstitution inflammatory syndrome.

The CPE rank is intended to be a practical strategy to estimate the CNS effectiveness of ART drug regimens taken by PLWH. There are several limitations, which have been evaluated between the CPE score and neurocognitive outcomes, resulting in inconclusive data and variations in the from studies, which used the same cART regimens in the CSF of PLWH. In summary, there are multifactorial causes behind CNS drug penetration [112]. Lately, more data have been included in the CPE rank as the levels of a determinate drug, in previous clinical studies, could improve neurocognitive impairment, i.e., its effectiveness in the CNS. Although the validity of suppression of HIV levels in CSF as a surrogate for novel treatments of the brain is unknown, substantial evidence supports it [112]. Data from important studies suggest that the CNS effectiveness of ART regimens could be considerably improved for a significant proportion of those receiving ART. Important challenges to alleviate neurologic damage of disease could include ameliorated therapies to treat infection of HIV-1 in the brain [115].

## 4. Conclusions

HAND is a debilitating and devastating complication of HIV infection. Success with cART continues to improve life expectancy, but with increasing age, manifestations of neurocognitive disorder in HIV/AIDS patients are increasing. Since HIV-1 protease inhibitors are an important component of ART regimens, targeting the CNS sanctuary of HIV with new protease inhibitors capable of crossing the BBB and interrupting HIV in the brain may be key to treating or even preventing the more severe form of HAND: HAD. Darunavir is an FDA-approved protease inhibitor widely used in PLWH, with a significantly greater virological response and immunological benefits compared to the standard of care. Protease inhibitors possess desirable features as drugs suitable for treating PLWH with wild-type and/or multidrug-resistant HIV-1 variants.

In addition, it is important to know that, although HAD is notable and this dementia has decreased substantially, most PLWH with HAND have a state of stability. Abnormalities in the CNS persist in PLWH even with systemic viral suppression. Several biomarkers of inflammation (neopterin) and markers of active axonal or neuronal injury and oxidative stress, among others, remain elevated in cART-treated PLWH with HAND. The lack of solutions for HAND in PLWH on therapy reveals a requirement for adjunctive approaches that go beyond the effects of cART. Novel studies demonstrated that a poloxamer-PLGA nano-formulation loaded with elvitegravir (EVG), a commonly utilized ART drug, is an efficient delivery method for EVG. The PLGA-EVG nanoparticles showed a positive stable state and a significant transmigration across an in vitro BBB model. PLGA-EVG nanoparticles indicated enhanced elimination of the virus in macrophages infected by HIV-1 after crossing the BBB model [116]. Another example using in vitro models used for the BBB in HIV infections in the CNS was postulated by Kodidela, S. et al., showing that cucurbitacin-D reduces HIV replication directly as well as across the BBB models. It is also effective against cigarette smoke condensate-induced HIV replication. This study provides the potential for cucurbitacin-D to be developed as an adjuvant therapy in HIV treatment. It may be utilized not only to eliminate HIV in the CNS but also to diminish the toxicity within the CNS of currently existing ART drugs [117]. These in vitro models include a subset of circulating immune cells, such as T cells and macrophages. Two of the major cellular reservoirs of latent HIV-1 in the brain are microglia and macrophages.

Furthermore, another important topic is the direct effects of HIV on neurons that are produced by proteins of the virus, induced by cells with HIV-1, and can damage neurons. However, focus should also be placed on indirect effects when neuronal cells are injured not only by the contact with viral proteins but also by inflammatory toxicity induced by astrocytic cells and microglia.

Finally, the existence of different clinical histories such as methamphetamine abuse is correlated with enhanced HIV-1 replication, increased gp120- and Tat-mediated neurotoxicity, and cognitive damage. In general, use of illegal substances is considered a potential potentiator of HAND. Promising therapeutic developments are being evaluated to find clinically approved treatments.

## 5. Future Directions

Infection of the CNS can be caused by the capability of HIV to cross the BBB. Infection of this compartment involves macrophages, which have a critical function in the neurodegeneration event, releasing inflammatory cytokines, proteins of HIV-1, and neurotoxins.

There are some studies that have demonstrated an evidence connection between microbial translocation and systemic immune stimulation, and this evidences that macrophage and microglia stimulation is related to the evolution of HIV disease. It has been evidenced that increased LPS levels, with high levels of macrophage and microglia activation, represent interrelated risk factors for HAND. Subsequently, bacterial products induce different signaling routes of pro- and anti-inflammatory TLRs, and the equity between them may affect the evolution of HAND. Novel methods to downmodulate LPS-induced macrophage stimulation could provide new strategies for anti-HIV and HAND therapy. Additionally, abolishing this immune activation occurs in brain compartments since the CNS remains one of the principal barriers to HIV-1 eradication and can block infection evolution beyond that restricted by ART.

Cells within the CNS have a restricted ability to replace and remove many types of cells with HIV-1 from the CNS, which may cause neurocognitive impairment and lead to bad results in PLWH. It is necessary to study novel markers applicable to monitoring reactivation or eradication of infected cells by the virus, securely and without difficulty. The total characterization of the CNS reservoir and the evolution of suitable markers to quantitate and monitor the CNS reservoir are important to the efficacious and secure elimination of HIV-1.

There are still gaps in this topic, and thus we should concentrate future efforts on further studies that shed light on this. Defining improved strategies for analyzing and evaluating HIV-1 repositories in the CNS and assessing the connections between drugs, brain penetration, and diffusion of drugs crossing the BBB and into the brain will be important points of reference in the effort to eradicate viral populations and permanent reservoirs throughout the body.

## Figures and Tables

**Figure 1 microorganisms-09-02537-f001:**
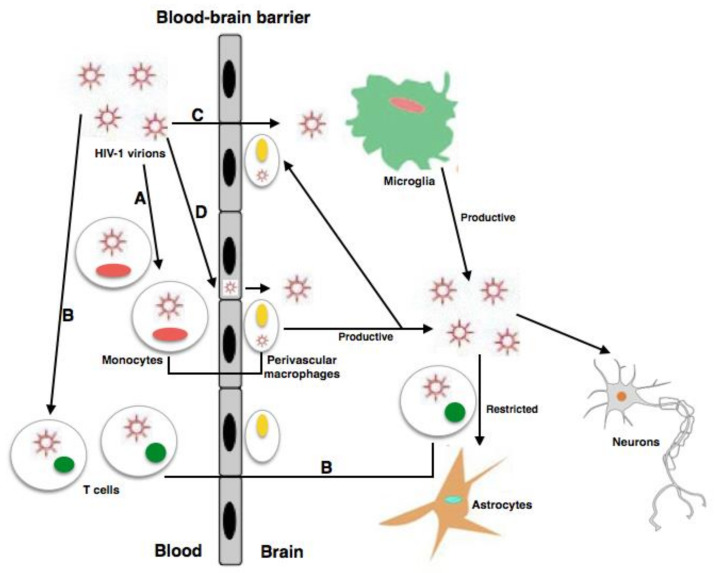
The mechanisms of HIV infection within the CNS. (A, B, C) HIV-1 enters the CNS through diverse pathways: (A) the Trojan horse process through which HIV-1-infected monocytes pass across the BBB and differentiate into perivascular macrophages; (B) the transfer into the CNS of HIV-1-infected CD4+ T cells; (C) entry into brain is possible in a direct way provided their is raised permeability, owing to dysfunctions and/or modified tissue. (D) Microglia, neurons, and astrocytes are the CNS-resident cells vulnerable to HIV-1 infection. Cell activation plays an important role in the release of proinflammatory cytokines and can increase changes in and the permeability of the BBB, thus helping the development of neuro-invasion of HIV and other viruses.

**Figure 2 microorganisms-09-02537-f002:**
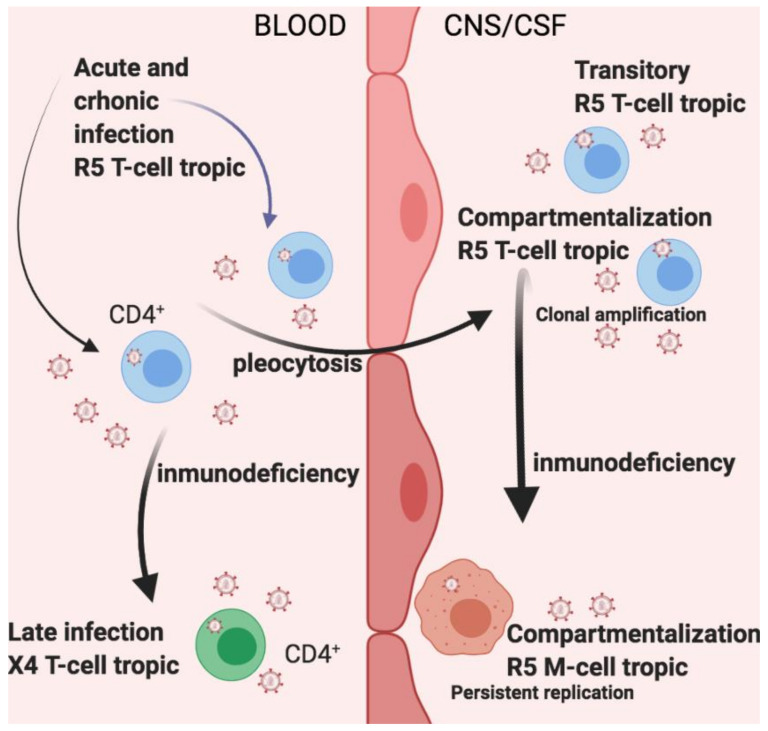
Infections, acute and chronic, by T-tropic CCR5-using (R5) viruses proceed in the systemic circulation. Populations of macrophage (M)-tropic HIV-1, which do not need high CD4 levels to infect cells and can proliferate within the brain in macrophages and other cells, are the second and more pathogenic type of compartmentalized infection related to HAD. Independent evolution happens at the systemic level with the development of CXCR4-using (X4) viruses; this is related to low blood CD4 cells and rapid advancement and may be shown in CSF.

**Table 1 microorganisms-09-02537-t001:** CNS penetration effectiveness score (CPE score).

Antiretroviral Drug Class	4 (Very Good)	3 (Good)	2 (Fair)	1 (Poor)
Nucleoside Reverse Transcriptase Inhibitors (NRTIs)	Zidovudine	Abacavir Emtricitabine	Didanosine Lamivudine Stavudine	Adefovir Tenofovir Zalcitabine
Non-Nucleoside Reverse Transcriptase Inhibitors (NNRTIs)	Nevirapine	Delavirdine Efavirenz	Etravirine	
Protease Inhibitors (PIs)	Amprenavir-r Indinavir-r	Amprenavir Danmavir Darunavir Fosamprenavir-r Indinavir Lopinavir-r	Atazanavir Atazanarir-r Fosamprenavir	Nelfinavir Ritonavir Saquinavir Saquinavir-r Tipranavir-r
Integrase Inhibitors		Elvitegravir Raltegravir		
Entry Inhibitors		Maraviroc Vicriviroc		Enfuvirfide T-1249

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
