# Peer review of "Crucial Role of Central Nervous System as a Viral Anatomical Compartment for HIV-1 Infection"

_microorganisms, 2021, doi:10.3390/microorganisms9122537_

Round 1

Reviewer 1 Report

Ana et al. review recent literature describing HIV infection of the central nervous system. This is a digestible, informative, and interesting review covering two major subtopics: (1) the problem of HIV-associated neurological disorder (HAND) and (2) how the CNS may serve as a long-term reservoir for life-long infection. I have corrections and suggestions for improvement or clarification but, overall, I liked the review.

  1. Overall, English is pretty fluid but the text will definitely benefit from a additional rounds of editing (typos, language, grammar, and capitalization issues).
  2. Abstract- HIV can infect the CNS but does not always does so, correct? Check language here.
  3. Typically cART is "combined" ART, not "current"?
  4. Section 2.1, P3: please expand on what is known regarding the mechanisms by which paroxetine, marviroc, and intranasal insulin are thought to be benefiting cognitive function. Interesting topic, and leaves the reader wanting more.
  5. Section 2.2, P1- Please define the relevant cognitive domains (i.e., go into the sufficient detail needed by a broad audience).
  6. Figure 1 proposes T cell-mediated delivery of HIV to the CNS as well as a “direct access” model wherein virions directly cross the BBB. However, neither is discussed in the text. The authors should do so.
  7. Section 2.4: While there is a significant amount of data on gp120, Tat, Vpr, Nef, etc. affecting cell responses and cytotoxicity; it seems really unclear as to what levels of these proteins are actually present in the CNS (either intra- or extracellularly) during the relevant phases of disease development with or without ART. Please discuss this issue, because at present it is really hard to gauge what might be real versus ex vivo artifacts or off-target effects from non-physiological concentrations of said products.
  8. Section 2.4. There are also recent studies of viral RNA triggering inflammatory cytokine production in monocytes and microglia (e.g, 33298546). Seems highly relevant and should be added to the review.
  9. Wasn’t clear to me if “HAD” vs. “HAND” was being used interchangeably or not, might review this carefully?
  10. Section 2.5, P4- Please expand on the immunoproteasome topic- not clear to me how the virus would affect this pathway? Would seem more reasonable that some drugs might affect it through, in particular protease inhibitors.
  11. Section 2.5, P5: I found this section on microbial products interesting but quite confusing as written. Please first explain to the reader how the microflora and general translocation is affected by HIV and/or therapy that might lead to these products access in the brain. Please define what the relevant elements are in more detail, and what they do (e.g., LPS, neuropod, TLRs, CX3CR1 cells, etc.).
  12. Section 3, P2: There is extensive discussion of how HIV might cross the BBB barrier to enter the brain, but not enough regarding how it might get out to re-establish a systemic infection in the context of a CNS reservoir.
  13. Section 3. P4: Just a minor detail- studies of SIV provide evidence for how SIV works, not HIV.
  14. Section 3 on compartmentalization would really benefit from a figure. Will be difficult for a broad audience to envision where these activities are happening in the brain and connections between the nervous system and blood and lymph.
  15. Section 3.2, P1: Section states that lamivudine, stavudine, and zidovudine decrease CNS cells- is this in general or are they able to selectively target infected cells? Please expand on selectivity/targeting as this is such an important issue for drugs in the brain.
  16. Section 3.3., P1: How is the CPE scoring actually done? Please include these details.

Author Response

Ana et al. review recent literature describing HIV infection of the central nervous system. This is a digestible, informative, and interesting review covering two major subtopics: (1) the problem of HIV-associated neurological disorder (HAND) and (2) how the CNS may serve as a long-term reservoir for life-long infection. I have corrections and suggestions for improvement or clarification but, overall, I liked the review.

    Overall, English is pretty fluid but the text will definitely benefit from a additional rounds of editing (typos, language, grammar, and capitalization issues).
Thank you for your very careful review of our paper, and for the comments, corrections and suggestions that ensued. We believe the paper has been significantly improved. In the final version of the manuscript, we detail the changes that you have suggested. We have included additional round of editing in all text.

    Abstract- HIV can infect the CNS but does not always does so, correct? Check language here.
    Typically cART is "combined" ART, not “current"?
According with the clever Reviewer’s comment, we have corrected this errors.
Modified text:
Abstract Human immunodeficiency virus 1 (HIV-1) establishes a chronic infection, which can lead to severe immunodeficiency CD4+ T cell despite reduction in HIV-1 ribonucleic acid (RNA) levels and increased survival of people living with HIV-1 by the treatment with combined antiretroviral therapies (cART). HIV-1 enters the central nervous system (CNS) and infects perivascular macrophages and microglia. Infection of the CNS produces neurological syndromes such as HIV-associated neurocognitive disorders (HAND). HAND contribute to important morbidity and mortality globally despite progress in HIV treatment through cART. The pathogenesis and the role of inflammation in HAND are still incompletely understood. Principally, growing evidences shows that the CNS may serve as an anatomical reservoir for viral replication and that the CNS compartmentalization has also been associated with the development of neurocognitive impairment, which has major implications for HIV eradication strategies. In this review, important concepts to understanding HAND and neuropathogenesis as well as the viral proteins involved in the CNS as an anatomical reservoir for HIV infection have been discussed. In addition, an overview of the current progress towards the therapeutic strategies for treatment of HAND has been reviewed. Further neurological research is needed to address neurological complications in people living with HIV (PLWH), particularly regarding CNS viral reservoirs and their effects on eradication.

 Section 2.1, P3: please expand on what is known regarding the mechanisms by which paroxetine, marviroc, and intranasal insulin are thought to be benefiting cognitive function. Interesting topic, and leaves the reader wanting more.
In accordance with reviewer’s suggest, we have added more information about this interesting topic.
Additional text:
“There is a important study that provides evidences that paroxetine reduced neuronal cell death induced by HIV-1 proteins in vitro and in vivo33. It has been showed that paroxetine prevented Tat-induced iNOS and inflammatory cytokine expression, and Ca2+-induced mitochondrial swelling, and attenuated the KCl-induced calcium responses. Also, data of this work suggests that the neuroprotective properties of paroxetine in neuronal cell death (seen in the brains of patients with HAND) are seen at concentrations of 0.5 to10 μM. Similar concentrations are also needed to enhance proliferation of neural progenitor cells. These concentrations can be easily achieved in the brain at therapeutic dosages since these compounds are preferentially taken up and enriched in brain tissue with 10–20 fold higher concentrations in the human brain compared to plasma33. Steady state levels are achieved over several weeks to months33. This is also supported by in vivo studies in which paroxetine showed neuroprotection and proliferation of neural progenitor cells following systemic administration. Therefore, treatment with paroxetine can be an effective neurorestorative approach for ameliorating neuronal death as well as defective neurogenesis in HIV-infected individuals with neurologic dysfunction33.
Maraviroc has anti-inflammatory properties, good antiretroviral efficacy in cells including those of MDM lineage and good CNS penetration34, highlighting its clear potential for therapeutic use. There is a recent study, provides the most convincing data supporting maraviroc intensification to date. By employing a randomized-controlled design, longer follow-up period, and optimal neuropsychological methods for longitudinal research, this study supports and extends recent observations of improved neurocognition of HIV-infected participants with some degree of cognitive impairment who underwent maraviroc intensification for 24 weeks34. That study also reported partial reversal of monocyte-mediated pathological changes previously associated with neurocognitive impairment, namely reducing the proportion of circling intermediate and nonclassical CD16-expressing monocytes, CD14+ HIV DNA monocyte burden and pro-inflammatory biomarker sCD163 levels in plasma34. 
The mechanisms responsible for the neuronal injury observed here are unknown. Kim et al. 2019 have shown that the restoration of dendritic arbors and memory after insulin treatment indicates that relevant neurons survive and can return to full functionality, similar to reversal of memory and synaptic deficits in amyloid beta precursor protein (APP) transgenic mice treated with insulin-like growth factor-2 (IGF-2)35 and transient loss of microtubule-associated protein 2 (MAP2) dendritic staining after moderate traumatic brain injury35. The proposed mechanism of dendritic structural stability includes reversible MAP2-microtubule association under the control of mitogen-activated protein kinase/extracellular signal-regulated kinase (MEK-ERK) and CaMKII pathways35. Notably, CaMKII and its regulator NRGN are transcriptionally downregulated in the infection of this work, as is the neurotrophin BDNF that enhances dendrite formation35, and they are restored to normal expression with insulin treatment.” (page 3; paragraph 4-page 4, paragraph 1)
Additional references:
33 Steiner JP, Bachani M, Wolfson-Stofko B, et al. Interaction of paroxetine with mitochondrial proteins mediates neuroprotection [published correction appears in Neurotherapeutics. 2016 Jan;13(1):237. Wang, Tonguang [Corrected to Wang, Tongguang]]. Neurotherapeutics. 2015;12(1):200-216. doi:10.1007/s13311-014-0315-9
34 Gates TM, Cysique, LA, Siefried KJ, Chaganti J, Moffat KJ, Brew BJ. Maraviroc-intensified combined antiretroviral therapy improves cognition in virally suppressed HIV-associated neurocognitive disorder, AIDS: February 20, 2016 - Volume 30 - Issue 4 - p 591-600 doi: 10.1097/QAD.0000000000000951 
35Kim BH, Kelschenbach J, Borjabad A, Hadas E, He H, Potash MJ,et al. Intranasal insulin therapy reverses hippocampal dendritic injury and cognitive impairment in a model of HIV-associated neurocognitive disorders in EcoHIV-infected mice. AIDS. 2019 May 1;33(6):973-984. doi: 10.1097/QAD.0000000000002150.

    Section 2.2, P1- Please define the relevant cognitive domains (i.e., go into the sufficient detail needed by a broad audience).
We have followed the reviewer’s recommendation and have defined, with additional specific    information, the relevant cognitive domains.
Text added:
“the cognitive impairment does not meet criteria for delirium or dementia and there is no evidence of another preexisting cause for the ANI7. 
MND represents a cognitive impairment, involves at least two cognitive domains that produces mild interference in daily function; self-report or observation by knowledgeable others of reduced mental acuity, inefficiency in work, homemaking, or social functioning of the PLWH and the cognitive impairment does not meet criteria for delirium or dementia7.
HAD is related to marked cognitive impairment, involves at least two cognitive domains that substantially interferes with daily functioning7; the pattern of cognitive impairment does not meet criteria for delirium (e.g., clouding of consciousness is not a prominent feature); or, if delirium is present, criteria for dementia need to have been met on a prior examination when delirium was not present. There is no evidence of another, preexisting cause for the dementia (e.g., other CNS infection, CNS neoplasm, cerebrovascular disease, preexisting neurologic disease, or severe substance abuse compatible with CNS disorder). If the individual with suspected HAD also satisfies criteria for a severe episode of major depression with significant functional limitations or psychotic features, or substance dependence, the diagnosis of HAD should be deferred to a subsequent examination conducted at a time when the major depression has remitted or at least 1 month has elapsed following cessation of substance use36.”

    Section 2.4: While there is a significant amount of data on gp120, Tat, Vpr, Nef, etc. affecting cell responses and cytotoxicity; it seems really unclear as to what levels of these proteins are actually present in the CNS (either intra- or extracellularly) during the relevant phases of disease development with or without ART. Please discuss this issue, because at present it is really hard to gauge what might be real versus ex vivo artifacts or off-target effects from non-physiological concentrations of said products.
We have added new text (page 7; last paragraph-page 8; first paragraph) in the paper answering the smart Reviewer’s inquiries. Also, additional references have been added in the manuscript 
Additional text:
There are studies that postulate that the causes of HIV-1 disease progression are multifaceted as multiple events occur leading to a cascade of pathological consequences. It is an attractive objective to correlate the observed increase in ROS formation with the augmented viral load burden within the CNS, although the leading causes might be various. Additionally, patients with late-stage HIV-1 infection have been shown to benefit from treatment with antioxidant compounds, as these remedies lengthen life expectancy but do not halt the ongoing oxidation phenomena, either because the process is irreversible or because of an excessive accumulation of ROS, which may no longer be scavenged by the weakened antioxidant cellular defense mechanisms. Additionally, a specific work said that because the extracellular Vpr concentrations that are very low in the early phases of disease compared with late stages, it is appealing, although speculative, to hypothesize that in early disease stages exogenous levels of Vpr are too low to cause immediate detrimental effects, and the observed downstream effects are generated only later during disease progression when Vpr levels rise above this “threshold.” It is possible, however, that in early stages, astrocytes, although exposed to subthreshold concentrations of Vpr, might become sensitized to subsequent exposure to Vpr so that lower amounts of Vpr are necessary to impair astrocytic functionality70. (page 7; last paragraph-page 8; first paragraph)
Additional reference:
70 Ferrucci A, Nonnemacher MR, Wigdahl B. Human immunodeficiency virus viral protein R as an extracellular protein in neuropathogenesis. Adv Virus Res. 2011;81:165-199. doi:10.1016/B978-0-12-385885-6.00010-9

    Section 2.4. There are also recent studies of viral RNA triggering inflammatory cytokine production in monocytes and microglia (e.g, 33298546). Seems highly relevant and should be added to the review.
We have followed the useful Reviewer’s recommendation and have added additional information about viral RNA triggering inflammatory cytokine production in monocytes and microglia.
Text added:
“Akiyama et al. 2020 have shown that HIV icRNA expression alone induces IFN-I-dependent proinflammatory responses in MDMs, even though HIV icRNA expression does not lead to production of new virions or functional viral proteins, including gp120 and Vpr71. A infection with virus expressing a Rev-mutant deficient for HIV icRNA nuclear export, which fails to induce innate immune activation in microglia, expresses multiply spliced viral RNAs, including those encoding Tat, suggesting that de novo Tat expression is not the trigger for HIV-induced microglia activation. Interestingly, HIV icRNA (gag mRNA) has been detected in the CSF from HIV-1+ individuals on cART71, and a highly sensitive RNAScope assay has revealed the presence of a significant number of SIV gag mRNA (icRNA)-positive cells in the brain of cART-suppressed monkeys71. Also, these viral icRNA-expressing cells in the brain, which are most likely microglia, induce proinflammatory cytokines and affect neuronal health in cART-suppressed individuals. Several drug candidates that suppress expression or stability of HIV icRNA, such as Tat and Rev inhibitors71, might have clinical benefit for suppressing HIV icRNA-induced aberrant inflammation and incidence of HAND in cART-suppressed patients71.” (page 8; second paragraph).
Additional reference:
71 Akiyama H, Jalloh S, Park S, Lei M, Mostoslavsky G, Gummuluru S. Expression of HIV-1 Intron-Containing RNA in Microglia Induces Inflammatory Responses. J Virol. 2020 Dec 9;95(5):e01386-20. doi: 10.1128/JVI.01386-20. Epub ahead of print. 

    Wasn’t clear to me if “HAD” vs. “HAND” was being used interchangeably or not, might review this carefully?
In accordance with accurate Reviewer’s recommendation, we have revised this error in all text (in red in the final version of the manuscript).

    Section 2.5, P4- Please expand on the immunoproteasome topic- not clear to me how the virus would affect this pathway? Would seem more reasonable that some drugs might affect it through, in particular protease inhibitors.
    Section 2.5, P5: I found this section on microbial products interesting but quite confusing as written. Please first explain to the reader how the microflora and general translocation is affected by HIV and/or therapy that might lead to these products access in the brain. Please define what the relevant elements are in more detail, and what they do (e.g., LPS, neuropod, TLRs, CX3CR1 cells, etc.).
According with the intelligent Reviewer’s comments, we have added and explained these pivotal topics.
Text added:
“It has been reported that the important component of the immunoproteasome, TRIM5α, was individually silenced using siRNA and the effects on IFNα-induced viral suppression were determined. The functional inter-dependence of the IFNα-activated anti-HIV-1 phenotypes of human TRIM5α and the immunoproteasome was established by the substantial diminution in the level of rescue from IFNα inhibition that was observed following PA28A silencing in cells that lacked TRIM5α75.Though there have been sporadic reports of human TRIM5α affecting HIV-1 infection either by suppression of certain HLA-associated cytotoxic T lymphocyte (CTL) escape mutant viruses, or by inducing autophagy in Langerhans cells, these findings demonstrate broad, non-strain specific inhibition of HIV-1 infection by human TRIM5α. Importantly, this report has reported that TRIM5α function is operative in CD4+ T cells, and is dependent on IFNα and activation of the immunoproteasome. Given that IFN levels are elevated during the acute and chronic phases of natural HIV-1 infection, it can be supposed that TRIM5α contributes to the immune control of HIV-1 in infected humans; a conclusion consistent with noted associations between favourable clinical outcomes and elevated TRIM5α expression or specific TRIM5α alleles75. 
Also, an interesting topic faces the possible effects of HIV-1 protease inhibitors on immunosurveillance probably will depend on the particular role of proteasome-dependent as opposed to proteasome-independent CTL epitope presentation as well as on the relative contributions of the CTL response as opposed to the recovery of CD4+ lymphocytes and antibody production76 in immune surveillance. Drugs with proteasome-inhibitory capacity as ritonavir/saquinavir, is able to modulate the presentation of Ags to CTLs and may perhaps be exploited further for the treatment of autoimmune disease, chronic immunopathologies, or disease caused by transplantation reactions.
On the other hand, another postulated mechanism for the sustained CNS inflammation is microglial priming from circulating microbial translocation products derived from gut bacteria and a disturbed microbiome. There is evidences that microbial products can transverse to the BBB through different mechanisms. It has also been suggested that the CNS inflammation in cART-treated individuals could be an attenuated form of immune reconstitution inflammatory syndrome77.
Microbial translocation has been found to be a major driver of morbidity and mortality in HIV infection, likely due to the persistent inflammation it induces and sustains78,79. Importantly, the associations between microbial translocation and disease progression and mortality are independent of whether the HIV-infected individual is virally suppressed with ART80. A commonly identified translocating microbial product is lipopolysaccharide (LPS) from the surface of Gram-negative bacteria. Several studies have demonstrated direct correlations between plasma LPS levels in HIV-infected individuals residual viremia, cellular activation including CD38+HLA-DR+ CD8+ T cells and activation of monocytes, interferon responsive genes such as MxA, and proinflammatory cytokines including IFN-α, IL-6, TNFα. In addition, LPS levels and/or bacterial DNA levels directly correlate with other markers of microbial translocation and innate immune activation such as soluble CD14 (sCD14; released by monocytes in response to bacterial stimulation), LPS-binding protein (LBP) and endotoxin. While it is unclear how much inflammation during HIV infection is directly attributed to microbial translocation given the many inflammatory mechanisms which occur during HIV infection (including virus replication, opportunistic infections, etc.), studies in the absence of HIV demonstrate relationships between microbial translocation and inflammation. In idiopathic CD4 lymphocytopenia (ICL), LPS is elevated and associated with proliferating CD4+ T cells, and colon LPS levels in uninfected pigtail macaques correlate with interferon responsive gene MxA in the GI tract80, demonstrating that microbial products can directly stimulate inflammatory responses. 
A recent study demonstrate that loss of immune stimulation by the gut microbiota leads to failure to control viral replication within the CNS leading to enhanced neuropathy. Thus, loss of protective resident microbes can lead to CNS dysfunction79.
There are different molecules that could influence the maturation of the CNS. LPS have been studied to be sufficient to prime microglia for antigen presentation to effectively clear virus79. While microglia have long been known to express Toll Like Receptors (TLRs), this family of receptors has been primarily thought to function only during infection. It has been showed that the microbiota regulates microglia function through TLR4, priming these cells to respond to infection. Microglia develop early in embryogenesis from yolk sac progenitors; however, in contrast to macrophages, microglia are long-lived without any significant input from circulating blood cells79. Also, there is evidence that gut microbial products are found circulating within the blood and could reach the CNS through this route81. There are data that demonstrate that TLR4 signaling by microglia is, in part, responsible for orchestrating microglia activation and the gut microbiota signals can be transmitted to the CNS from the enteric nervous system75. Indeed, enteroendocrine cells have been reported to contain neuropods (axon-like basal process that contains neurofilaments, which are typical structural proteins of axons) that are directly linked to neuronal cells and are able to transmit signals to the CNS81. Brown et al. administered LPS orally and this administration limits effects to gastrointestinal exposure, but oronasalpharyngeal and pneumonic exposure may be occurring as well. Relatedly, differences observed between feeding mice LPS alone, and in combination with a TLR1/2 ligand (Pam3CysK4) suggest that more research could be performed investigating the interaction between multiple TLR ligands on microglia function and response to infection. Moreover, in this study cannot completely rule out a contribution from gut-resident CX3CR1 cells to this phenotype or other migrating DC populations82. Recent works have indicated that cells from the gut can migrate to the brain and there exist a population of long-lived CX3CR1 cells within the gut83. Other studies postulate that disruption of the gut epithelial barrier may permit the unregulated translocation of gut microbes into the lamina propria. Thus, bacterial factors can infiltrate the gut-associated lymphoid tissues (GALT), and the blood lumen, where they interact with various immune cells and can stimulate effector-type T-cell differentiation84 Regulatory T-cells that survey the GALT, blood, and CSF and changes to the local microbiome can promote T-cell brain infiltration. Circulating bacterial factors can upregulate inflammatory cytokine levels, affect BBB integrity and promote neuroinflammation80 .Also, LPSs are produced by bacterial factors and can act on endothelial TLRs to promote neuroinflammation and CNS disease84,85.” (page 8; last paragraph-page 10; first paragraph).
Additional bibliography:
75 Jimenez-Guardeño JM, Apolonia L, Betancor G, Malim MH. Immunoproteasome activation enables human TRIM5α restriction of HIV-1. Nat Microbiol. 2019;4(6):933-940. doi:10.1038/s41564-019-0402-0
76 André P, Groettrup M, Klenerman P, et al. An inhibitor of HIV-1 protease modulates proteasome activity, antigen presentation, and T cell responses. Proc Natl Acad Sci U S A. 1998;95(22):13120-13124. doi:10.1073/pnas.95.22.13120
80 Zevin AS, McKinnon L, Burgener A, Klatt NR. Microbial translocation and microbiome dysbiosis in HIV-associated immune activation. Curr Opin HIV AIDS. 2016;11(2):182-190. doi:10.1097/COH.0000000000000234

    Section 3, P2: There is extensive discussion of how HIV might cross the BBB barrier to enter the brain, but not enough regarding how it might get out to re-establish a systemic infection in the context of a CNS reservoir.
In accordance with accurate Reviewer’s recommendation, I believe that this topic is very important but that it deserves a broader description and approach that we couldn’t dedicate to it in this paper and it could be a great idea for future studies and future manuscripts that will focus on the subject. Thank you very much for this contribution and we will take this genial idea to the next work.

    Section 3. P4: Just a minor detail- studies of SIV provide evidence for how SIV works, not HIV.
In accordance with accurate Reviewer’s recommendation, we have revised this detail in all text.

    Section 3 on compartmentalization would really benefit from a figure. Will be difficult for a broad audience to envision where these activities are happening in the brain and connections between the nervous system and blood and lymph.
In accordance with Reviewer’s advice, we made include additional Figure 2 (page 12) to improve this section.

    Section 3.2, P1: Section states that lamivudine, stavudine, and zidovudine decrease CNS cells- is this in general or are they able to selectively target infected cells? Please expand on selectivity/targeting as this is such an important issue for drugs in the brain.
We absolutely agree with the Reviewer’s  and have included important information about this pivotal issue.
Text added:
“Lamivudine, stavudine and zidovudine are used in Neuro-cART regimens, but there are studies that indicate that these drugs may not target all the susceptible HIV-1 target cell populations in the CNS, with potential implications for their inclusion in Neuro-cART regimens. 
Astrocytes are critical for maintaining normal brain homeostasis and astrocyte dysfunction is known to contribute to HIV-1 neuropathogenesis but these drugs could potentially lead to astrocyte infection remaining integrated, which may contribute to neurocognitive impairment despite virological suppression in plasma. HIV-1 infection of astrocytes is predominantly restricted to the expression of genes encoding the regulatory/accessory HIV-1 proteins112, some of which are neurotoxic (for example the HIV-1 Tat protein)108, and contributes to the persistent viral reservoir within the brain. These drugs have markedly reduced effectiveness in astrocytes compared to macrophages, the underlying mechanism for this remains unknown. Three possible explanations could address the reduced effectiveness of these drugs in astrocytes; differences in cellular uptake of NRTIs, inefficient or incomplete drug activation (due to lower levels of cellular kinases or competition with the natural substrates for the kinases), and inefficient incorporation into DNA (due to higher levels of endogenous nucleotides)112. (page 12; paragraph 3)

    Section 3.3., P1: How is the CPE scoring actually done? Please include these details.
We have added additional text in accordance with clever Reviewer’s proposal.
Text added:
“The CPE rank is intended to be a practical approach to estimate the CNS effectiveness of ART drug regimens taken by PLWH. An inherent limitation of the CPE rank method is the paucity of available information on drug penetration into the CNS. Because drug concentrations and virologic suppression cannot be directly measured in brain tissue, surrogate markers are used instead, such as chemical characteristics and CSF drug levels114. However, the availability of data on ARV drug levels in CSF varies, significantly limiting the ability to make direct comparisons between different drugs. To deal with these limitations in available data, it has been incorporated, an additional class of information into the CPE rank: the degree to which a drug, in previous clinical studies, improved neurocognitive impairment, ie, its effectiveness in the CNS. Although the validity of suppression of HIV levels in CSF as a surrogate for treatment of the brain is unknown, considerable evidence supports it114. Data of important studies suggest that the CNS effectiveness of ART regimens could be substantially improved for a significant proportion of those receiving ART therapy. Improving the treatment of CNS HIV infection can benefit neurological outcomes and reduce overall disability due to neurocognitive impairment in HIV infection115”.

Reviewer 2 Report

This review article by Borrajo et al. is well-organized and thorough, summarizing knowledge and raising questions regarding the Central Nervous System (CNS), HIV infections and HIV-Associated Neurocognitive Disorders (HAND). Overall, the review is clearly written and organized. The authors outline aspects of cells permissive for HIV infection in the CNS as well as the role of inflammation and potentially, LPS in the CNS and its role in HAND. The evolution of HIV in the CNS is considered, as well as the CNS as a reservoir for latent HIV, and the authors focus on this aspect of HIV infection of the CNS. Borrajo et al. discuss the CNS compartmentalization and evolution of HIV in the CNS, which the authors argue, should be considered a distinct anatomical site for HIV evolution. The models presented and the discussion of which cART is effective in the CNS is also discussed and is quite informative for the readership. 

This reviewer supports acceptance with some modifications. In many instances, there is awkward English usage that needs to be significantly improved. Some of these are noted in the specific comments section, below. There needs to be clarification of some scientific concepts, as outlined in the specific comments; improvement in quality of the Table; some reference modification; and some additions to the conclusion section. See Specific Comments below.

Specific Comments

Title: “Crucial Role of Central Nervous System as a viral anatomical for HIV-1 infection”. What does this mean? As a “viral anatomical”? Is something missing here? Also, either all words need to be capitalized or only the first and “HIV” would be capitalized. Do you mean as a distinct viral anatomical compartment?

Names in Last, First Name would be reversed (i.e., Ana Barrajo not Borrajo Ana).

There were no line numbers or page numbers provided with the manuscript I was asked to review. Therefore, I can only reference sections. Not having page or line numbers makes it very difficult to comment.

Abstract:

“…to severe immunodeficiency”. This needs to be more specific about what kind of immunodeficiency it is that occurs – T cell deficiency? This statement comes across as vague. While HIV infections are known to have an immunodeficiency component, there is also hyperactivation of the immune system, so the aspects of immunodeficiency that occurs needs to be spelled out. Also, with cART one would need to spell out whether there is this immunodeficiency in these cases.

“Anti-“ not uppercase.

“cART” c is not for ‘current’, but ‘combination’. Change that.

There are a lot of inappropriately uppercase words.

“Sets up” comes across as jargon. Please change, i.e., to something like “infects perivascular…”

“invades” instead, “accesses” or “enters”

“…still incompletely knowledgeable” Not correct language – change to “are unknown”, or “incompletely understood” .

“in this review, important concepts in the field of HAND as the neuropathogenesis, HIV viral proteins involved and CNS as viral reservoir for HIV infection have been discussed.”

It doesn’t make sense to say, “the field of HAND as…” This could be changed perhaps to “understanding HAND and neuropathogenesis as well as the viral proteins involved in the CNS as an anatomical reservoir for HIV.

Typo - “adittion” to addition

Introduction

Introduction repeats the first line of the Abstract. These should not be exactly the same, so change this sentence. Add what immunodeficiency you are referring to, specifically, here (if not added in the abstract).

Again, “c” is for combination, not current. This cannot be the same as the Abstract

There needs to be distinction when referencing papers reporting on Non-human primate models for HIV vs. HIV infection in humans. The references for this ending statement: “for prolonged periods of time” states it is the case for HIV and does not specify SIV.

Here, you have reference 2 about SIV models. You need to clarify when you are talking about NHP models and when you are taking about human infection. There are differences and confusing the two is not accurate. Here you should say something like, REF 1 in Human and (REF2) NHP models of HIV-1 infection, such as SIVmac.

 How are MDM difference from other macrophage (i.e., non-monocyte-derived macrophage), when referring to in vivo cell types? Is this a term you are using to reflect the in vitro models? Please clarify. Are MDM distinct from dendritic cell origin-macrophage? This cell type should be clarified and comes across as a term for an in vitro model being used vs. in vivo phenomena. Without definition and clarification, the ontogeny of this cell type is unclear.

“HIV-associated neurological disease provokes important morbidity and mortality globally and can be due to HIV replication, opportunistic infections, or comorbidities.” Strange wording. Good point.

the viral reservoir or reservoir(s)…”

Instead of “patients”, use either PLWH - People living with HIV; or HIV infected persons/people.

“viral reservoir still persists causing neurocognitive disorders” Is it absolutely clear that the viral reservoirs are what is causing the problem? Do these 2 reviews that are cited clearly outline that information?

“Taking into account the restrictions of current therapeutic strategies, novel approaches are a key point in the neuro-HIV research. Part of these approaches involve gene therapy, gene editing, RNA interference, and modulation of different cellular physiological processes23.”

This (above) seems to me like a weak point if it is just citing one review. Also, when citing reviews, it is preferable to write “reviewed in (REFERENCE)”

“Even though the precise etiology of HIV-Associated Neurocognitive Disorders (HAND)…’

Question: If it has not been “deeply” studied, then how do you know it is due to the viral reservoir persistence?

  1. HAND

 Weird English “ 2.1. Advances and evolution of HIV infection and their connection to the antiretroviral treatment.”

Overall, it is clear this is not written by an English speaker. There are excessive instances of unclear language that is structurally incorrect, and the science is unclear due to this language usage in many cases.

 The first paragraph is good and the points are clear.

“…and it is necessary managing…” ? This language is unclear and confusing. It is not clear what is meant here.  

“In this light, it is important providing new insights” Unclear language.

No page numbers, so difficult to reference in this review.

“ARVs.” Switch to this nomenclature? If so, define here.

“Currently, no HAND-specific therapies exist, but small trials of paroxetine and maraviroc showed some benefit in improving neurocognitive function in HIV+ cART-treated adults while trials of intranasal insulin are ongoing after in vitro evidence suggested insulin may have neuroprotective effects in HIV infection32.

(For above; again, difficult as there are no line numbers) I would add an introductory line on talking about the specific cARTs – or refer to the table here for definitions of the classes of ART these are.

2.2 OK

2.3

“procedure”? Process is a better word.

“1-2 weeks later virus come into the systemic circulation” This step is unclear due to the language. Virus shed from the CNS? Or for CNS entry from the blood?

“accentuating”? Maybe “supporting” or “permissive for”  

I would define excitoxic for those not familiar with brain/neuronal terminology.

 2.4

Define “NMDARs” within the text. Even if the glossary, the acronym needs to be defined within the text.

2.5

“on the development of HAD.: Here, the subtitle is HAND, but using “HAD.” In this paragraph discern the difference between HAND and HAD if you are focusing on HAD. Restate this distinction in this section.

 “…inflammatory responses, triggered by HIV, stimulate the proteasome to evolve into an ‘immunoproteasome’ that obstructs the turnover of folded proteins in brain cells and affects cellular homeostasis and response to stress73” That is a good point and interesting. Is there more than one reference for this hypothesis?

“On the other hand, another postulated mechanism for the sustained CNS inflammation is microglial priming”

Is there evidence that LPS and other microbial products translocate into the brain? If so, how, i.e., bound to what receptors/ligands/cells?

OK, this is answered in future paragraphs, but it could be stated in a sentence in the earlier paragraph that there is evidence microbial products can transverse the BBB through  a number of mechanisms…

  1. The CNS As A Viral Reservoir For Hiv Persistence

“Furthermore, specific viral proteins (including Tat, Rev and Nef) can be originated” Originated is not the word, but “produced”is preferable.

3.1

“HAD shelter different word such as “harbor” would be preferred.

 (EC50) = effective concentration, not efficacious concentration. Please update.

3.2. Towards strategies for the eradication of CNS reservoirs.

This section has good points, and it is informative.

 3.3. CNS Penetration-Effectiveness (CPE) scoring system

Table 1. This is a good Table, but it has very poor resolution quality – this needs to be improved.

“these researches” “This research” - replace all of the instances.

  1. Conclusions

The Conclusions do not seem like conclusions, but suitable for “Future Directions”.

Conclusions should include one or two lines per section of the review to summarize conclusions from each of those sections. Then, I would include these points.

I would recommend writing a summary of each subsection before these “future directions” and in the current conclusion section. I would include the summary to discuss how some cART does not access the BBB; how this is evaluated; weighing out evidence for direct vs. indirect models for HAND with HIV infections; and the cell types mostly responsible for productive infection in brain; age and other comorbidities; HIV evolution and the argument for brain as separate compartment of HIV evolution.

It would be useful to add other in vitro models used for BBB in HIV infections in the CNS and comment on these. One such example is Gong, Y. et al. Novel Elvitegravir Nanoformulation for Drug Delivery across the Blood-Brain Barrier to Achieve HIV-1 Suppression in the CNS Macrophages. Sci. Rep. 2020, 10, 3835. Another example using this model was published in Kodidela, S. et al. Anti-HIV Activity of Cucurbitacin-D against Cigarette Smoke Condensate-Induced HIV Replication in the U1 Macrophages. Viruses 2021, 13, 1004.

Author Response

This review article by Borrajo et al. is well-organized and thorough, summarizing knowledge and raising questions regarding the Central Nervous System (CNS), HIV infections and HIV-Associated Neurocognitive Disorders (HAND). Overall, the review is clearly written and organized. The authors outline aspects of cells permissive for HIV infection in the CNS as well as the role of inflammation and potentially, LPS in the CNS and its role in HAND. The evolution of HIV in the CNS is considered, as well as the CNS as a reservoir for latent HIV, and the authors focus on this aspect of HIV infection of the CNS. Borrajo et al. discuss the CNS compartmentalization and evolution of HIV in the CNS, which the authors argue, should be considered a distinct anatomical site for HIV evolution. The models presented and the discussion of which cART is effective in the CNS is also discussed and is quite informative for the readership. 

This reviewer supports acceptance with some modifications. In many instances, there is awkward English usage that needs to be significantly improved. Some of these are noted in the specific comments section, below. There needs to be clarification of some scientific concepts, as outlined in the specific comments; improvement in quality of the Table; some reference modification; and some additions to the conclusion section. See Specific Comments below.
Thank you for your comments and suggestions that allowed us to greatly improve the quality of the manuscript. We agree with all your comments, and as you proposed, we have try to modified all the important points that you have suggested. 

Specific Comments

Title: “Crucial Role of Central Nervous System as a viral anatomical for HIV-1 infection”. What does this mean? As a “viral anatomical”? Is something missing here? Also, either all words need to be capitalized or only the first and “HIV” would be capitalized. Do you mean as a distinct viral anatomical compartment?

Names in Last, First Name would be reversed (i.e., Ana Barrajo not Borrajo Ana).

There were no line numbers or page numbers provided with the manuscript I was asked to review. Therefore, I can only reference sections. Not having page or line numbers makes it very difficult to comment.

Abstract:

“…to severe immunodeficiency”. This needs to be more specific about what kind of immunodeficiency it is that occurs – T cell deficiency? This statement comes across as vague. While HIV infections are known to have an immunodeficiency component, there is also hyperactivation of the immune system, so the aspects of immunodeficiency that occurs needs to be spelled out. Also, with cART one would need to spell out whether there is this immunodeficiency in these cases.

“Anti-“ not uppercase.

“cART” c is not for ‘current’, but ‘combination’. Change that.

There are a lot of inappropriately uppercase words.

“Sets up” comes across as jargon. Please change, i.e., to something like “infects perivascular…”

“invades” instead, “accesses” or “enters”

“…still incompletely knowledgeable” Not correct language – change to “are unknown”, or “incompletely understood” .

“in this review, important concepts in the field of HAND as the neuropathogenesis, HIV viral proteins involved and CNS as viral reservoir for HIV infection have been discussed.”

It doesn’t make sense to say, “the field of HAND as…” This could be changed perhaps to “understanding HAND and neuropathogenesis as well as the viral proteins involved in the CNS as an anatomical reservoir for HIV.

Typo - “adittion” to addition
We have followed the useful Reviewer’s recommendation and have done all the modifications suggested in the Title and Abstract. These modifications appear in red in the text.

Crucial role of central nervous system as a viral anatomical compartment for HIV-1 infection
Ana Borrajo1,2, Valentina Svicher1, Romina Salpini1, Michele Pellegrino3 and Stefano Aquaria
Abstract Human immunodeficiency virus 1 (HIV-1) establishes a chronic infection, which can lead to severe immunodeficiency CD4+ T cell despite reduction in HIV-1 ribonucleic acid (RNA) levels and increased survival of people living with HIV-1 by the treatment with combined antiretroviral therapies (cART). HIV-1 enters the central nervous system (CNS) and infects perivascular macrophages and microglia. Infection of the CNS produces neurological syndromes such as HIV-associated neurocognitive disorders (HAND). HAND contribute to important morbidity and mortality globally despite progress in HIV treatment through cART. The pathogenesis and the role of inflammation in HAND are still incompletely understood. Principally, growing evidences shows that the CNS may serve as an anatomical reservoir for viral replication and that the CNS compartmentalization has also been associated with the development of neurocognitive impairment, which has major implications for HIV eradication strategies. In this review, important concepts to understanding HAND and neuropathogenesis as well as the viral proteins involved in the CNS as an anatomical reservoir for HIV infection have been discussed. In addition, an overview of the current progress towards the therapeutic strategies for treatment of HAND has been reviewed. Further neurological research is needed to address neurological complications in people living with HIV (PLWH), particularly regarding CNS viral reservoirs and their effects on eradication.
Introduction

Introduction repeats the first line of the Abstract. These should not be exactly the same, so change this sentence. Add what immunodeficiency you are referring to, specifically, here (if not added in the abstract).

According with the intelligent Reviewer’s comments, we have modified and explained this unclear sentences adding other phrases instead of the errated sentences and correct references, errors and typo errors.
Test added: 
“HIV-1 infection is marked by a progressive depletion of peripheral CD4+ T cells, T-cell dysfunction, thymic dysfunction, lymphoid destruction, and pan-cellular defects attributed to stem cell dysfunction. In the majority of HIV-infected persons, the ultimate result of this immune dysfunction is development of the opportunistic infections and malignancies associated with AIDS”.

Again, “c” is for combination, not current. This cannot be the same as the Abstract.
Corrected.

There needs to be distinction when referencing papers reporting on Non-human primate models for HIV vs. HIV infection in humans. The references for this ending statement: “for prolonged periods of time” states it is the case for HIV and does not specify SIV.
Here, you have reference 2 about SIV models. You need to clarify when you are talking about NHP models and when you are taking about human infection. There are differences and confusing the two is not accurate. Here you should say something like, REF 1 in Human and (REF2) NHP models of HIV-1 infection, such as SIVmac.
Test added: 
“(this data has been shown in a recent and important study with SIV-infected Asian macaques)” (page 2; paragraph 2).
Referenced corrected.

 How are MDM difference from other macrophage (i.e., non-monocyte-derived macrophage), when referring to in vivo cell types? Is this a term you are using to reflect the in vitro models? Please clarify. Are MDM distinct from dendritic cell origin-macrophage? This cell type should be clarified and comes across as a term for an in vitro model being used vs. in vivo phenomena. Without definition and clarification, the ontogeny of this cell type is unclear.
Test added:
“It is know that macrophages have different origins: embryonic yolk sac derived, fetal liver derived, and/or bone marrow MDMs. MDMs are a cell type generated from peripheral blood monocytes and are widely used to model macrophages for in vitro studies. Considering the important role of both tissue-resident macrophages and MDMs in homeostasis and disease it has always been key to develop representative in vitro models to study these cells.” (page 1; paragraph 2)

“HIV-associated neurological disease provokes important morbidity and mortality globally and can be due to HIV replication, opportunistic infections, or comorbidities.” Strange wording. Good point.

“ the viral reservoir or reservoir(s)…”

Instead of “patients”, use either PLWH - People living with HIV; or HIV infected persons/people.

“viral reservoir still persists causing neurocognitive disorders” Is it absolutely clear that the viral reservoirs are what is causing the problem? Do these 2 reviews that are cited clearly outline that information?

“Taking into account the restrictions of current therapeutic strategies, novel approaches are a key point in the neuro-HIV research. Part of these approaches involve gene therapy, gene editing, RNA interference, and modulation of different cellular physiological processes23.”

This (above) seems to me like a weak point if it is just citing one review. Also, when citing reviews, it is preferable to write “reviewed in (REFERENCE)”

“Even though the precise etiology of HIV-Associated Neurocognitive Disorders (HAND)…’
All these modification have been done.
Text modified:
“HIV-associated neurological disease can produce important morbidity and mortality globally and can be due to HIV replication, opportunistic infections, or comorbidities. Hypertension, diabetes, and vascular disease are comorbidities in older HIV-infected patients15 potentially affecting brain structure and function16. HIV can primarily or secondarily affect all parts of the nervous system, including the brain, meninges, spinal cord, nerve roots, peripheral nerves, and muscles17. This infection status often causes neurological symptoms including cognitive impairment and motor disturbance7. In general, this disorder is characterized by a combination of virus-related neurological disorders and neuronal-tissue inflammation7,18. The use of cART with improved ability of penetration of the blood-brain barrier have drastically reduced the incidence of these complications18,19. Nevertheless, since not all anti-HIV drugs are capable to cross the BBB with high effectiveness, viral reservoirs still persist causing neurocognitive disorders20. As a result, these clinical symptoms still remain an important problem for people living with HIV (PLWH) particularly for those who are in special conditions like children or patients with low adherence to treatment21 and are increasingly common in the HIV population with advancing age since this is a risk factor associated with functional deterioration and disability22. Other important risk factors are low CD4+ T-cells, the increase of plasma viral load, Hepatitis C Virus (HCV)-coinfection and metabolic comorbidities in HIV-neurocognitive infected PLWH7,19. Taking into account the restrictions of current therapeutic strategies, novel approaches are a key point in the neuro-HIV research. Part of these approaches involve gene therapy, gene editing, RNA interference, and modulation of different cellular physological processes (reviewed in Ojha et al. 2017 and Kwarteng et al. 2017)23,24. Even though the precise etiology of HIV-Associated Neurocognitive Disorders (HAND) in the cART era has not been still deeply studied, chronic and persistent inflammation in CNS is a common feature of HIV infection that may lead to accumulation of neurological damage25.” (page 2; paragraph 8)

References added:
20. Saylor D, Dickens AM, Sacktor N, Haughey N, Slusher B, Pletnikov M, et al. HIV-associated neurocognitive disorder--pathogenesis and prospects for treatment. Nat Rev Neurol. 2016 Apr;12(4):234-48. doi: 10.1038/nrneurol.2016.27. Epub 2016 Mar 11. Review. Erratum in: Nat Rev Neurol. 2016 May;12(5):309. 
24. Kwarteng A, Ahuno ST, Kwakye-Nuako G. The therapeutic landscape of HIV-1 via genome editing. AIDS Res Ther. 2017;14(1):Published 2017 Jul 14. doi:10.1186/s12981-017-0157-8
25Marban C, Forouzanfar F, Ait-Ammar A, et al. Targeting the Brain Reservoirs: Toward an HIV Cure. Front Immunol. 2016;7:397. Published 2016 Sep 30. doi:10.3389/fimmu.2016.00397

Question: If it has not been “deeply” studied, then how do you know it is due to the viral reservoir persistence?
Thanks for the clever question. Although, it has not been “deeply” studied, there are different studies that detail the evidence for early brain infection and the brain as a sanctuary for HIV, as well as considering how and why inflammation is sustained in chronic HIV infection, even when systemic virological control is achieved20
Also, in other work it have been studied that HAND remains a common cause of cognitive impairment and has persisted even in individuals who have received CART. In addition, early HIV infection of the CNS is believed to contribute to the development of HAND, and evidence suggests that the CNS can subsequently serve as a reservoir for ongoing HIV replication, thereby limiting the opportunity for a sterilizing cure or eradication24

    HAND

 Weird English “ 2.1. Advances and evolution of HIV infection and their connection to the antiretroviral treatment.”

Overall, it is clear this is not written by an English speaker. There are excessive instances of unclear language that is structurally incorrect, and the science is unclear due to this language usage in many cases.

 The first paragraph is good and the points are clear.

“…and it is necessary managing…” ? This language is unclear and confusing. It is not clear what is meant here.  

“In this light, it is important providing new insights” Unclear language.

No page numbers, so difficult to reference in this review.

“ARVs.” Switch to this nomenclature? If so, define here.

“Currently, no HAND-specific therapies exist, but small trials of paroxetine and maraviroc showed some benefit in improving neurocognitive function in HIV+ cART-treated adults while trials of intranasal insulin are ongoing after in vitro evidence suggested insulin may have neuroprotective effects in HIV infection32.

(For above; again, difficult as there are no line numbers) I would add an introductory line on talking about the specific cARTs – or refer to the table here for definitions of the classes of ART these are.

2.2 OK

2.3

“procedure”? Process is a better word.

“1-2 weeks later virus come into the systemic circulation” This step is unclear due to the language. Virus shed from the CNS? Or for CNS entry from the blood?

“accentuating”? Maybe “supporting” or “permissive for”  
In accordance with accurate Reviewer’s recommendations, we have corrected in the text (in red in the final version of the manuscript) all these errors. Sorry for forget to insert the page numbers. 
Changed text:
“2. HAND 
2.1. Advances and evolution of the antiretroviral treatment to HIV infection.
cART have improved the treatment of HIV infection, changing the life expectancy of PLWH with access to treatment. As a consequence, immune reconstitution and opportunistic diseases have become infrequent, nevertheless cognitive disorders in PLWH still persist20. 
Neurologic complications of HIV infection are unfortunately common, even in the era of effective cART. The successful virologic control has extended the life span of PLWH but it is necessary the management of long-term complications of HIV disease, such as neurocognitive disorders and peripheral neuropathy26. The availability of robust and affordable virological and immunological markers can help in solving these issues by providing information on the burden of HIV-1 reservoir in all the anatomical compartments in which the virus replicates as well as on persistent inflammation, immune activation and senescence despite successful virological suppression26. For all these reasons, it is important providing new insights in evaluation and monitoring of HIV-1 infection from a virological, immunological and clinical perspective26. Particular attention has been focused on role of novel parameters (such as total HIV-1 DNA, residual viremia, and immunological markers) in optimizing treatment strategies, enhancing medical adherence, and individualizing monitoring26.
It is important to note that the impact of HAND is likely to continue to grow as the global HIV+ population ages as mounting evidence suggests that HIV exacerbates age-associated cognitive decline27 and older age is associated with increased risk of HAND16,28 Older HIV+ individuals also demonstrate greater than expected brain atrophy on neuroimaging studies which was associated with impaired performance in multiple cognitive domains compared to older HIV- individuals29. Longitudinal studies demonstrate synergistic effects of HIV and aging on cognitive function30. Cognitive decline is likely multifactorial due to direct damage from the virus and indirect damage through secondary risk factors, including vascular disease, chronic drug use, and toxic long-term effects of antiretroviral drugs. Importantly, premature age-associated neurocognitive decline correlated with structural and functional brain changes on neuroimaging and histopathology, is observed in some PLWH at younger ages than would otherwise be expected and may be related to accelerated aging as discussed above31. Currently, no HAND-specific therapies exist, but small trials of paroxetine and maraviroc showed some benefit in improving neurocognitive function in HIV+ cART-treated adults while trials of intranasal insulin are ongoing after in vitro evidence suggested insulin may have neuroprotective effects in HIV infection32 (Table 1). 
There is a important study that provides evidences that paroxetine reduced neuronal cell death induced by HIV-1 proteins in vitro and in vivo33. It has been showed that paroxetine prevented Tat-induced iNOS and inflammatory cytokine expression, and Ca2+-induced mitochondrial swelling, and attenuated the KCl-induced calcium responses. Also, data of this work suggests that the neuroprotective properties of paroxetine in neuronal cell death (seen in the brains of patients with HAND) are seen at concentrations of 0.5 to10 μM. Similar concentrations are also needed to enhance proliferation of neural progenitor cells. These concentrations can be easily achieved in the brain at therapeutic dosages since these compounds are preferentially taken up and enriched in brain tissue with 10–20 fold higher concentrations in the human brain compared to plasma33. Steady state levels are achieved over several weeks to months33. This is also supported by in vivo studies in which paroxetine showed neuroprotection and proliferation of neural progenitor cells following systemic administration. Therefore, treatment with paroxetine can be an effective neurorestorative approach for ameliorating neuronal death as well as defective neurogenesis in HIV-infected individuals with neurologic dysfunction33.
Maraviroc has anti-inflammatory properties, good antiretroviral efficacy in cells including those of MDM lineage and good CNS penetration34, highlighting its clear potential for therapeutic use. There is a recent study, provides the most convincing data supporting maraviroc intensification to date. By employing a randomized-controlled design, longer follow-up period, and optimal neuropsychological methods for longitudinal research, this study supports and extends recent observations of improved neurocognition of HIV-infected participants with some degree of cognitive impairment who underwent maraviroc intensification for 24 weeks34. That study also reported partial reversal of monocyte-mediated pathological changes previously associated with neurocognitive impairment, namely reducing the proportion of circling intermediate and nonclassical CD16-expressing monocytes, CD14+ HIV DNA monocyte burden and pro-inflammatory biomarker sCD163 levels in plasma34. 
The mechanisms responsible for the neuronal injury observed here are unknown. Kim et al. 2019 have shown that the restoration of dendritic arbors and memory after insulin treatment indicates that relevant neurons survive and can return to full functionality, similar to reversal of memory and synaptic deficits in amyloid beta precursor protein (APP) transgenic mice treated with insulin-like growth factor-2 (IGF-2)35 and transient loss of microtubule-associated protein 2 (MAP2) dendritic staining after moderate traumatic brain injury35. The proposed mechanism of dendritic structural stability includes reversible MAP2-microtubule association under the control of mitogen-activated protein kinase/extracellular signal-regulated kinase (MEK-ERK) and CaMKII pathways35. Notably, CaMKII and its regulator NRGN are transcriptionally downregulated in the infection of this work, as is the neurotrophin BDNF that enhances dendrite formation35, and they are restored to normal expression with insulin treatment. 
Development of validated biomarkers and improved clinical neurocognitive tests that can holistically and accurately assess the risk of developing HAND are also needed to facilitate future trials of novel HAND therapies.
Also, particular attention has been focused on role of novel parameters (such as total HIV-1 DNA, residual viremia, and immunological markers) in optimizing treatment strategies, enhancing medical adherence, and individualizing monitoring26.
2.2. Classification of HAND
Several classification systems have been suggested during the time for the diagnosis of HAND. The Frascati criteria36 are the most broadly used and are considered the gold standard for the diagnosis of HAND. Its scheme identifies three severity levels of HAND: Asymptomatic Neurocognitive Impairments (ANI), Mild Neurocognitive Disorders (MND), and HIV-Associated Dementia (HAD)7. ANI is typified by cognitive impairment, involves at least two cognitive domains that does not interfere with everyday function; the cognitive impairment does not meet criteria for delirium or dementia and there is no evidence of another preexisting cause for the ANI7. 
MND represents a cognitive impairment, involves at least two cognitive domains that produces mild interference in daily function; self-report or observation by knowledgeable others of reduced mental acuity, inefficiency in work, homemaking, or social functioning of the PLWH and the cognitive impairment does not meet criteria for delirium or dementia7.
HAD is related to marked cognitive impairment, involves at least two cognitive domains that substantially interferes with daily functioning7; the pattern of cognitive impairment does not meet criteria for delirium (e.g., clouding of consciousness is not a prominent feature); or, if delirium is present, criteria for dementia need to have been met on a prior examination when delirium was not present. There is no evidence of another, preexisting cause for the dementia (e.g., other CNS infection, CNS neoplasm, cerebrovascular disease, preexisting neurologic disease, or severe substance abuse compatible with CNS disorder). If the individual with suspected HAD also satisfies criteria for a severe episode of major depression with significant functional limitations or psychotic features, or substance dependence, the diagnosis of HAD should be deferred to a subsequent examination conducted at a time when the major depression has remitted or at least 1 month has elapsed following cessation of substance use36.
2.3. Pathology of NeuroAIDS
The pathogenesis of HAND itself is complicated and multidimensional and it has been suggested in recent studies that the pathophysiology of HAND is more likely to be associated with functional alterations in neurons. 
Productive HIV infection occurs in perivascular macrophages, MDM and microglia34. While it is well consolidated that neuronal injury and loss are correlated with the evolution of HAND manifestations, paucity of information is available on HIV capability to infect neurons13, 35. In accordance with the broadly accepted model, HIV penetrates into the CNS within 1–2 weeks of systemic infection, through a “Trojan Horse” process36 (Figure 1), crossing the BBB throughout infected monocytes that, subsequently, differentiate into macrophages35. Infection and activation of cells takes place via direct contact with infected cells  (“Trojan horse” cells such as perivascular macrophages, astrocytes, and microglia)37. Although astrocytes are sensitive to HIV infection, they do not promote productive infection, conversely, perivascular macrophages and microglia are the only cells in the CNS able of HIV supporting infection in the brain13,36 (Figure 1). In the setting of HAND, symptoms are correlated with loss of neurons and cellular damage. Furthermore, previous studies have shown that chemokines, cytokines and viral proteins released from infected and/or activated microglia/macrophages and astrocytes are presumably responsible for neuronal injury38. The controversial question is whether or not HIV can infect neurons, even at low levels, and a lot of studies, dating back to the 1980s and 90s, report the ability of HIV to infect neurons in the brain in vivo38. Previous works using in situ PCR and immunohistochemistry informed about the existence of HIV genetic material and antigens in neurons, respectively38. Further researches isolating neurons from post-mortem brain tissues of PLWH utilizing Laser Capture Microdissection (LCM) described the presence of HIV proviral DNA in neurons by PCR39,40. Previous work, employed hyperbranched multidisplacement techniques for whole-gene amplification via PCR and analyzed the existence of HIV DNA in single neurons collected by LCM from autopsy brain tissue41. In vitro studies reported that human neuronal cell lines could be infected with HIV42. Nevertheless, verification of pathologically significant infection of human neurons in vivo is still lacking. In addition to its presence in the CNS of adult PLWH, HIV is also detected in the CNS of infected pediatric patients, as well as in the developing fetal brain43.” (Page 3; first paragraph-Page 5 first paragraph)

I would define excitoxic for those not familiar with brain/neuronal terminology.
Text added: 
“It’s important to highlight that excitotoxicity is a complex process triggered by glutamate receptor activation that results in the degeneration of dendrites and cell death. Normal amounts of glutamate receptor activation can damage neurons under conditions of metabolic and oxidative stress, which occur in neurodegenerative disorders.” (page 6; paragraph 4)

 2.4

Define “NMDARs” within the text. Even if the glossary, the acronym needs to be defined within the text.
Error corrected.

2.5

“on the development of HAD.: Here, the subtitle is HAND, but using “HAD.” In this paragraph discern the difference between HAND and HAD if you are focusing on HAD. Restate this distinction in this section.

Text modified with specific terminology
“Apart from the direct neurotoxicity of HIV proteins, mononuclear phagocytes, including perivascular macrophages and resident microglia have a significant impact on the development of most severe level of HAND: HAD.
Inflammation plays a key role in the production of events that lead to neurodegeneration in HIV infection. The introduction of HIV into the brain across the BBB is carried out by circulating monocytes, in response to the chemotactic signals expressed within the parenchyma, and the monocytes are responsible for the establishment of infection in perivascular macrophages of CNS41, microglia34 and astrocytes72 .
Zhou et al., have shown that the direct and effective infection of the CNS macrophages is crucial for HAD manifestation69. They have found very extensive HIV productive infection in PLWH with rapid progression of disease with relatively low cellular infiltration compared to those who progress more slowly69. Strong predilection of infected macrophages and CD8+ T cells was typical of the deeper midline and mesial temporal structures uniquely in HAD patients, which has some influence on neurocognitive impairment during HIV infection69.
One question that has not been thoroughly studied is why CNS inflammation is maintained even when the viral replication is suppressed by cART73. On the one hand, a hypothesis supposes that the inflammatory responses, triggered by HIV, stimulate the proteasome to evolve into an ‘immunoproteasome’ that obstructs the turnover of folded proteins in brain cells and affects cellular homeostasis and response to stress74, resulting in severe damaged neuronal and synaptic protein dynamics, in some way contributing to HAD. The result of a important previous work suggest that the inmunoproteasomes may be involved with the local regulation of synaptic proteins in HIV-1 infection. In turn, synaptosome protein changes associated with inmunoproteasomes induction could lead to morphological changes in the synaptodendritic arbor, which is correlated with HIV-1-associated neurocognitive impairment74.” (page 8; paragraph 2).

 “…inflammatory responses, triggered by HIV, stimulate the proteasome to evolve into an ‘immunoproteasome’ that obstructs the turnover of folded proteins in brain cells and affects cellular homeostasis and response to stress73” That is a good point and interesting. Is there more than one reference for this hypothesis?

“On the other hand, another postulated mechanism for the sustained CNS inflammation is microglial priming”

Is there evidence that LPS and other microbial products translocate into the brain? If so, how, i.e., bound to what receptors/ligands/cells?

OK, this is answered in future paragraphs, but it could be stated in a sentence in the earlier paragraph that there is evidence microbial products can transverse the BBB through  a number of mechanisms…
“It has been reported that the important component of the immunoproteasome, TRIM5α, was individually silenced using siRNA and the effects on IFNα-induced viral suppression were determined. The functional inter-dependence of the IFNα-activated anti-HIV-1 phenotypes of human TRIM5α and the immunoproteasome was established by the substantial diminution in the level of rescue from IFNα inhibition that was observed following PA28A silencing in cells that lacked TRIM5α75.Though there have been sporadic reports of human TRIM5α affecting HIV-1 infection either by suppression of certain HLA-associated cytotoxic T lymphocyte (CTL) escape mutant viruses, or by inducing autophagy in Langerhans cells, these findings demonstrate broad, non-strain specific inhibition of HIV-1 infection by human TRIM5α. Importantly, this report has reported that TRIM5α function is operative in CD4+ T cells, and is dependent on IFNα and activation of the immunoproteasome. Given that IFN levels are elevated during the acute and chronic phases of natural HIV-1 infection, it can be supposed that TRIM5α contributes to the immune control of HIV-1 in infected humans; a conclusion consistent with noted associations between favourable clinical outcomes and elevated TRIM5α expression or specific TRIM5α alleles75. 
Also, an interesting topic faces the possible effects of HIV-1 protease inhibitors on immunosurveillance probably will depend on the particular role of proteasome-dependent as opposed to proteasome-independent CTL epitope presentation as well as on the relative contributions of the CTL response as opposed to the recovery of CD4+ lymphocytes and antibody production76 in immune surveillance. Drugs with proteasome-inhibitory capacity as ritonavir/saquinavir, is able to modulate the presentation of Ags to CTLs and may perhaps be exploited further for the treatment of autoimmune disease, chronic immunopathologies, or disease caused by transplantation reactions.
On the other hand, another postulated mechanism for the sustained CNS inflammation is microglial priming from circulating microbial translocation products derived from gut bacteria and a disturbed microbiome. There is evidences that microbial products can transverse to the BBB through different mechanisms. It has also been suggested that the CNS inflammation in cART-treated individuals could be an attenuated form of immune reconstitution inflammatory syndrome77.
Microbial translocation has been found to be a major driver of morbidity and mortality in HIV infection, likely due to the persistent inflammation it induces and sustains78,79. Importantly, the associations between microbial translocation and disease progression and mortality are independent of whether the HIV-infected individual is virally suppressed with ART80. A commonly identified translocating microbial product is lipopolysaccharide (LPS) from the surface of Gram-negative bacteria. Several studies have demonstrated direct correlations between plasma LPS levels in HIV-infected individuals residual viremia, cellular activation including CD38+HLA-DR+ CD8+ T cells and activation of monocytes, interferon responsive genes such as MxA, and proinflammatory cytokines including IFN-α, IL-6, TNFα. In addition, LPS levels and/or bacterial DNA levels directly correlate with other markers of microbial translocation and innate immune activation such as soluble CD14 (sCD14; released by monocytes in response to bacterial stimulation), LPS-binding protein (LBP) and endotoxin. While it is unclear how much inflammation during HIV infection is directly attributed to microbial translocation given the many inflammatory mechanisms which occur during HIV infection (including virus replication, opportunistic infections, etc.), studies in the absence of HIV demonstrate relationships between microbial translocation and inflammation. In idiopathic CD4 lymphocytopenia (ICL), LPS is elevated and associated with proliferating CD4+ T cells, and colon LPS levels in uninfected pigtail macaques correlate with interferon responsive gene MxA in the GI tract80, demonstrating that microbial products can directly stimulate inflammatory responses. 
A recent study demonstrate that loss of immune stimulation by the gut microbiota leads to failure to control viral replication within the CNS leading to enhanced neuropathy. Thus, loss of protective resident microbes can lead to CNS dysfunction79.
There are different molecules that could influence the maturation of the CNS. LPS have been studied to be sufficient to prime microglia for antigen presentation to effectively clear virus79,80. While microglia have long been known to express Toll Like Receptors (TLRs), this family of receptors has been primarily thought to function only during infection. It has been showed that the microbiota regulates microglia function through TLR4, priming these cells to respond to infection. Microglia develop early in embryogenesis from yolk sac progenitors; however, in contrast to macrophages, microglia are long-lived without any significant input from circulating blood cells79. Also, there is evidence that gut microbial products are found circulating within the blood and could reach the CNS through this route81. There are data that demonstrate that TLR4 signaling by microglia is, in part, responsible for orchestrating microglia activation and the gut microbiota signals can be transmitted to the CNS from the enteric nervous system75. Indeed, enteroendocrine cells have been reported to contain neuropods (axon-like basal process that contains neurofilaments, which are typical structural proteins of axons) that are directly linked to neuronal cells and are able to transmit signals to the CNS81. Brown et al. administered LPS orally and this administration limits effects to gastrointestinal exposure, but oronasalpharyngeal and pneumonic exposure may be occurring as well. Relatedly, differences observed between feeding mice LPS alone, and in combination with a TLR1/2 ligand (Pam3CysK4) suggest that more research could be performed investigating the interaction between multiple TLR ligands on microglia function and response to infection. Moreover, in this study cannot completely rule out a contribution from gut-resident CX3CR1 cells to this phenotype or other migrating DC populations82. Recent works have indicated that cells from the gut can migrate to the brain and there exist a population of long-lived CX3CR1 cells within the gut83,84. Other studies postulate that disruption of the gut epithelial barrier may permit the unregulated translocation of gut microbes into the lamina propria. Thus, bacterial factors can infiltrate the gut-associated lymphoid tissues (GALT), and the blood lumen, where they interact with various immune cells and can stimulate effector-type T-cell differentiation Regulatory T-cells that survey the GALT, blood, and CSF and changes to the local microbiome can promote T-cell brain infiltration. Circulating bacterial factors can upregulate inflammatory cytokine levels, affect BBB integrity and promote neuroinflammation85 .Also, LPSs are produced by bacterial factors and can act on endothelial TLRs to promote neuroinflammation and CNS disease85.” (page 8; paragraph 6 to page 10; first paragraph)

References added:
70 Akiyama H, Jalloh S, Park S, Lei M, Mostoslavsky G, Gummuluru S. Expression of HIV-1 Intron-Containing RNA in Microglia Induces Inflammatory Responses. J Virol. 2020 Dec 9;95(5):e01386-20. doi: 10.1128/JVI.01386-20. Epub ahead of print. 
75 Jimenez-Guardeño JM, Apolonia L, Betancor G, Malim MH. Immunoproteasome activation enables human TRIM5α restriction of HIV-1. Nat Microbiol. 2019;4(6):933-940. doi:10.1038/s41564-019-0402-0
76 André P, Groettrup M, Klenerman P, et al. An inhibitor of HIV-1 protease modulates proteasome activity, antigen presentation, and T cell responses. Proc Natl Acad Sci U S A. 1998;95(22):13120-13124. doi:10.1073/pnas.95.22.13120
80 Zevin AS, McKinnon L, Burgener A, Klatt NR. Microbial translocation and microbiome dysbiosis in HIV-associated immune activation. Curr Opin HIV AIDS. 2016;11(2):182-190. doi:10.1097/COH.0000000000000234
85 Logsdon AF, Erickson MA, Rhea EM, Salameh TS, Banks WA. Gut reactions: How the blood-brain barrier connects the microbiome and the brain. Exp Biol Med (Maywood). 2018;243(2):159-165. doi:10.1177/1535370217743766 

    The CNS As A Viral Reservoir For Hiv Persistence

“Furthermore, specific viral proteins (including Tat, Rev and Nef) can be originated” Originated is not the word, but “produced”is preferable.

3.1

“HAD shelter different word such as “harbor” would be preferred.

 (EC50) = effective concentration, not efficacious concentration. Please update.

3.2. Towards strategies for the eradication of CNS reservoirs.

This section has good points, and it is informative.

 3.3. CNS Penetration-Effectiveness (CPE) scoring system

Table 1. This is a good Table, but it has very poor resolution quality – this needs to be improved.

“these researches” “This research” - replace all of the instances.
In accordance with Reviewer’s advice, all the error have been corrected in the text (in red) (page and we have improved the resolution of the Table 1.
Text modified:
“3. The CNS As A Viral Reservoir For HIV Persistence
The importance of the CNS as a reservoir for persistent HIV infection has gained importance since it is considered an immunologically privileged site that can dampen efforts focused on developing eradication strategies. The different criteria to consider the CNS as a viral reservoir include i) incorporating an integrated virus, ii) HIV- 1 maintained in a latent condition, iii) virus is capable of replicating within long-lived cells resident in CNS and is found at significant abundance82. Furthermore, specific viral proteins (including Tat, Rev and Nef) can be produced in the absence of virion generation90. 
BBB discriminates and limits the crossing of cell and macromolecules from the systemic circulation to the CNS34. In spite of the BBB, HIV-1 penetrates the brain and infects primarily microglia and macrophages, that transport and release HIV-1 into the brain and allow the passing of cell-free virus through of the CNS91 . BBB leads to differing mechanisms of immune surveillance in the CNS compared with the periphery.
Certainly, to achieve a real sterilizing cure, the latent HIV harbored in the brain should be eradicated since it can be reactivated and can then reseed a systemic infection. In vivo, HIV infection in the brain occurs in perivascular macrophages and microglia90. Previous studies have identified viral particles in the brains of PLWH with undetectable viral load in the blood and cerebrospinal fluid (CSF)91 . Moreover, it have shown that viruses in CNS have unique long terminal repeat (LTR) promoters, with mutations in the Sp motif directly adjacent of the two NF-κB binding sites, which fosters viral latency92. The perivascular macrophages, microglia and astrocytes (target cells in the CNS), are long half-lives that permit the virus to linger within CNS cells and allow the preservation of the CNS viral reservoir. At last, the recurrence of CNS cell infection has been studied in recent works that examined macrophages, microglia and astrocytes. PLWH with normal neurocognitively condition presented 17%, 14% and 11% infection of macrophages, microglia and astrocytes, respectively, as long as injured PLWH showed 30%, 9% and 19% infection, respectively93 . Then, other studies of infected brain tissue have also exposed that the frequency of HIV-1 infection is more elevated at or close to blood vessels than in regions apart from the vessel94 . 
Previous in vivo studies, based on simian immunodeficiency virus (SIV) models, have produced strong confirmation for the persistence of SIV DNA even after prolonged suppression of viral replication with cART95 and other studies showed the important role of macrophage populations for the productive infection by HIV, with the use of BrdU labelling and markers specific to stages of macrophage differentiation96. 
Numerous research works of HIV RNA levels in the CSF have also suggested the existence of latent infection in the brain despite plasma virus suppression below measureable clinical limits. This event is known as CSF viral escape and supports the role of a CNS reservoir97. This phenomenon can happen in as many as 5–10% of cART recipients and is associated with immune activation  and major depressive disorder98 and most likely originates from local viral reservoirs91. A better knowledge of CSF viral escape could contribute to obtain crucial insights about CNS HIV infection including the CNS cell types producing HIV during infection and could be key to the successful eradication of both latent and productive HIV from the brain92 . However, analysis of CSF viral escape are  restricted in all contexts by its relatively low prevalence and the need for a lumbar puncture for identification92.
3.1. Compartmentalization of HIV in the CNS
CNS compartmentalization develops when virus, penetrating into the CNS, undergoes divergent evolution from systemic virus resulting in genetically distinct viral strains in the CNS99. This process usually begins during primary HIV infection and has also been associated with the development of neurocognitive impairment100. 
HIV-1 in the CNS has been widely studied utilizing autopsic brain tissues or CSF. Researches of HIV-1 in the CNS using brain samples of patients collected at autopsy show that these patients with HAND harbor different viruses in their brains101 . The CSF can be utilized to evaluate viral dynamics and CNS viral populations in PLWH. The different HIV-1 populations in CSF versus blood have been studied, and these studies have proposed that viral populations in the CSF and blood can come from diverse pathways102(Figure 2). Three states of virus have been shown in the CSF when compared to the blood: equilibrated (where viral populations in the blood and CSF are analogous), compartmentalized (where virus from blood and CSF viral populations are very different, expressing independently evolution of populations in these compartments), and clonal amplification (where a single variant is largely extended within a compartment)103  (Figure 2).
Compartmentalized viral populations, genetically diverse from viral populations can be identified in the CSF/CNS during the whole of the progress of infection and the replication occurs in the periphery104 . Two important classifications of compartmentalization have been studied in previous works105. The first is clonal amplification, the fast growth of genetically very similar variants producing a CSF viral population of minor complexity. These viral populations demand elevated levels of CD4 for entry, and are CCR5 (R5)-using T cell-tropic106 (Figure 2). Clonally amplified populations have been discovered as early as two to six months post infection, during the primary infection107. Another more complicated classification of CSF viral compartmentalization has also been described, frequently made up of macrophage-tropic virus. These more genetically complex populations  frequently coincide with a prolonged time since HIV-1 infection, demonstrative of a more extensive stage of isolated replication and evolution of the population and entry phenotype108 . 
Compartmentalization that has also been studied during primary infection can happens in PLWH without obvious neurocognitive symptoms of HAND109 due to this the possibility of early detection of compartmentalized CSF variants is proposed to recognize PLWH with a elevated risk of developing HIV-associated neurocognitive complications. Sturdevant et al. have recently studied that a compartmentalized CSF/CNS viral lineage that have been settled less than six months post infection remained during a time of at least two years, exhibiting the perpetuation and development of a compartmentalized viral population within the CNS over a extensive period of time beginning during early infection110. Furthermore, in this work, in other subjects during primary infection, numerous compartmentalized populations from other PLWH in the primary infection, were identified within the CNS at diverse time points post infection, showing that even when compartmentalized populations are not preserved, the CNS may supply a permissive enviroment for viral replication111.
The importance of the adequate penetrance of diverse antiretrovirals into the brain is a subject of further studies112 and it has been fully corroborated that lower concentrations of antiretrovirals drugs in the CNS do not achieve optimal virological suppression. There are few pharmacological studies of cART with respect to tissue macrophages. A much elevated maximal effective concentration (EC50) in macrophages than in lymphocytes might diminish cART effectiveness in this cell type113. Also, a not insignificant percentage of astrocytes isolated from autopsy brain tissues of HIV+ patients include integrated HIV114. The capability of HIV-1 to integrate into terminally differentiated astrocytes indicate a stable reservoir of provirus in the CNS that affects to the evolution and the achievements of strategies involved in the eradication of HIV-141,114.
3.2. Towards strategies for the eradication of CNS reservoirs.
Numerous studies take on the existence of a unique viral reservoir in the CNS. The CNS has important challenges to eradication procedures. The first step is to get to target CNS reservoir during eradication strategies and identify the efficient drugs for the stimulation of HIV-1 in the non-CNS compartment and that can appropriately pass through the BBB91. Due to the CNS is an immune advantaged compartment, cART have not effective ingress through BBB and efficiency within the CNS115. Previously, lamivudine, stavudine and zidovudine acted by favoring the decrease of CNS cells, however, at present, others antiretroviral drugs like protease inhibitors are an important component of cART targeting the CNS sanctuary of HIV with new protease inhibitors capable of crossing BBB and interrupting HIV in the brain may be key to treating or even preventing HAND107. 
Lamivudine, stavudine and zidovudine are used in Neuro-cART regimens, but there are studies that indicate that these drugs may not target all the susceptible HIV-1 target cell populations in the CNS, with potential implications for their inclusion in Neuro-cART regimens. 
Astrocytes are critical for maintaining normal brain homeostasis and astrocyte dysfunction is known to contribute to HIV-1 neuropathogenesis but these drugs could potentially lead to astrocyte infection remaining integrated, which may contribute to neurocognitive impairment despite virological suppression in plasma. HIV-1 infection of astrocytes is predominantly restricted to the expression of genes encoding the regulatory/accessory HIV-1 proteins108, some of which are neurotoxic (for example the HIV-1 Tat protein)108, and contributes to the persistent viral reservoir within the brain. These drugs have markedly reduced effectiveness in astrocytes compared to macrophages, the underlying mechanism for this remains unknown. Three possible explanations could address the reduced effectiveness of these drugs in astrocytes; differences in cellular uptake of NRTIs, inefficient or incomplete drug activation (due to lower levels of cellular kinases or competition with the natural substrates for the kinases), and inefficient incorporation into DNA (due to higher levels of endogenous nucleotides)108. 
Darunavir is an FDA approved protease inhibitor widely used in treatment-experienced PLWH, with significantly greater virological response and immunological benefits compared to the standard care. Ghosh et al. have realized a structure-based modification resulted in very potent brain-penetrating protease inhibitors, GRL-04810, GRL-05010108. They have also designed other inhibitors such as GRL-0739 and GRL-10413 which show BBB efficacy. Data of these study suggest that GRL-04810, GRL-05010, GRL-0739 and GRL-10413 have several advantages: (a) they form extensive interactions with the residues in the active site of HIV-1 protease, leading to excellent antiviral activity against a broad spectrum of drug-resistant HIV-1 isolates and variants (b) the hydrophobic interactions and logD values indicate a very good lipophilicity profile and (c) they can efficiently penetrate the BBB and retain their activity108. Therefore, they possess desirable features as a drug suitable for treating patients infected with wild-type and/or multidrug-resistant HIV-1 variants. The current data warrant further consideration of these novel PIs for the treatment of HIV/AIDS. It is necessary to evaluate their other drug properties including their pharmacokinetics, pharmacodynamics, and oral bioavailability in the clinical setting108.
3.3. CNS Penetration-Effectiveness (CPE) scoring system 
Antiretroiral drugs have been classified in accordance with a CNS Penetration-Effectiveness (CPE) scoring system which is a scale from 1 to 4, with 1 being less advantageous and 4 the most helpful108 (Table 1). 
Active cART refers to cART regimens that have a minimum combined CPE score of 8 or more10. Hence, cART with a CPE score of at least 8 or more must be used to guarantee its effectiveness in the CNS and the impediment of the establishment of de-novo infection103,108 . Gray et al.  have exposed that viruses from different compartments have a distinct behavior and response to antiretroviral drugs, with brain viruses being less responsive than, for example, blood viruses109 . In the end, for the choice of an efficient ARV it is important to guarantee immunity and it is effective in the elimination of any cell that harbor reactivated virus. In contrast with other body organs, the CNS holds an modified immune system due to its immune-favored condition within the CNS91,115 . In summary, the set of this research show that a cure approach based on antiretroviral drugs, with CNS bioavailability and effectiveness to eradicate the CNS viral reservoirs and implicate the increase and promotion of the CNS immune system to help viral depuration, is necessary. Nevertheless, this would have to be adjusted with care to prevent, CNS immune reestablishment inflammatory syndrome.” (page 10; paragraph 4- page 13; third paragraph)

    Conclusions

The Conclusions do not seem like conclusions, but suitable for “Future Directions”.

Conclusions should include one or two lines per section of the review to summarize conclusions from each of those sections. Then, I would include these points.

I would recommend writing a summary of each subsection before these “future directions” and in the current conclusion section. I would include the summary to discuss how some cART does not access the BBB; how this is evaluated; weighing out evidence for direct vs. indirect models for HAND with HIV infections; and the cell types mostly responsible for productive infection in brain; age and other comorbidities; HIV evolution and the argument for brain as separate compartment of HIV evolution.

It would be useful to add other in vitro models used for BBB in HIV infections in the CNS and comment on these. One such example is Gong, Y. et al. Novel Elvitegravir Nanoformulation for Drug Delivery across the Blood-Brain Barrier to Achieve HIV-1 Suppression in the CNS Macrophages. Sci. Rep. 2020, 10, 3835. Another example using this model was published in Kodidela, S. et al. Anti-HIV Activity of Cucurbitacin-D against Cigarette Smoke Condensate-Induced HIV Replication in the U1 Macrophages. Viruses 2021, 13, 1004.
We absolutely agree with the Reviewer’s and have included these crucial works in our paper and have written a summary that include all sections suggest by the Reviewer.
Text addedt:
4. Conclusions
HAND is a debilitating to devastating complication of HIV infection. Success with combined cART continues to improve life expectancy, but with increasing age, manifestations of neurocognitive disorder for HIV/AIDS patients are increasing. Since HIV-1 protease inhibitors are an important component of ART regimens, targeting the CNS sanctuary of HIV with new protease inhibitors capable of crossing BBB and interrupting HIV in the brain may be key to treating or even preventing the form more severe of HAND: HAD. Darunavir is an FDA approved protease inhibitor widely used in PLWH, with significantly greater virological response and immunological benefits compared to the standard care. Protease inhibitors possess desirable features as a drug suitable for treating patients infected with wild-type and/or multidrug-resistant HIV-1 variants.
Also its important to know that HAD has significantly decreased with cART, HAND still persists in a large proportion of PLWH with well-controlled HIV, and substantial evidence suggests that the CNS remains abnormal even in the context of apparently successful systemic viral suppression. Many challenges remain in optimizing HIV treatment, and questions persist regarding cART strategies that may be neuroprotective or will most effectively ameliorate accrued neurologic disease. The incomplete resolution of HAND in patients on HIV treatment suggests a need for adjunctive therapies that go beyond the effects of cART. Novel studies demonstrated that a        poloxamer-PLGA nanoformulation loaded with elvitegravir (EVG), a commonly used antiretroviral drug, is an efficient delivery approach for EVG. The PLGA-EVG nanopartticles showed a favorable stability and safety profile and a significant transmigration across an in vitro BBB model. Most importantly, PLGA-EVG nanoparticles indicated an improved viral suppression in HIV-1-infected macrophages after crossing the BBB model116. Another example using in vitro models used for BBB in HIV infections in the CNS was postulated by Kodidela, S. et al. that showed that Cucurbitacin-D reduces HIV replication directly as well as across the BBB models. It is also effective against cigarette smoke condensate-induced HIV replication. This study provides the potential for Cucurbitacin-D to be developed as adjuvant therapy in HIV treatment. It may be used not only to suppress HIV in the brain, but also to reduce the CNS toxicity of currently existing ART drugs117. These models in vitro include a subset of circulating immune cells, including T cells and macrophages. Microglia and macrophages are the main cellular targets for HIV infection and productive viral replication in the CNS.
    Furthermore its a important topic, direct effects of HIV on neurons are produced by viral proteins (gp120, gp41, Tat, Nef, Rev and Vpr) that are released from infected cells and can damage neurons. However, indirect effects of HIV on neurons involve the activation of glial cells (both infected and uninfected) resulting in the production and release of neurotoxic cytokines.
   Finally, comorbidities ranging from aging to substance abuse have been implicated as potentiators of HAND. For example, the dopamine release associated with increased drug use may be associated with increased virus uptake in macrophages. Future studies will be critical to ensure optimal care for patients with diverse clinical histories.
5. Future directions” (page 14)-page 15, third paragraph).
